# Shed EBA-175 mediates red blood cell clustering that enhances malaria parasite growth and enables immune evasion

May M Paing[1†], Nichole D Salinas[1,2†], Yvonne Adams[3], Anna Oksman[4], Anja TR Jensen[3], Daniel E Goldberg[4], Niraj H Tolia[1,2]*

[1]Department of Molecular Microbiology, Washington University School of Medicine, St. Louis, United States; [2]Laboratory of Malaria Immunology and Vaccinology, National Institute of Allergy and Infectious Diseases, National Institutes of Health, Bethesda, United States; [3]Centre for Medical Parasitology at Department of Immunology and Microbiology (ISIM), Faculty of Health and Medical Sciences, University of Copenhagen, Copenhagen, Denmark; [4]Department of Medicine, Washington University School of Medicine, St. Louis, United States

**Abstract** Erythrocyte Binding Antigen of 175 kDa (EBA-175) has a well-defined role in binding to glycophorin A (GpA) during *Plasmodium falciparum* invasion of erythrocytes. However, EBA-175 is shed post invasion and a role for this shed protein has not been defined. We show that EBA-175 shed from parasites promotes clustering of RBCs, and EBA-175-dependent clusters occur in parasite culture. Region II of EBA-175 is sufficient for clustering RBCs in a GpA-dependent manner. These clusters are capable of forming under physiological flow conditions and across a range of concentrations. EBA-175-dependent RBC clustering provides daughter merozoites ready access to uninfected RBCs enhancing parasite growth. Clustering provides a general method to protect the invasion machinery from immune recognition and disruption as exemplified by protection from neutralizing antibodies that target AMA-1 and RH5. These findings provide a mechanistic framework for the role of shed proteins in RBC clustering, immune evasion, and malaria.
DOI: https://doi.org/10.7554/eLife.43224.001

*For correspondence:
niraj.tolia@nih.gov

[†]These authors contributed equally to this work

**Competing interests:** The authors declare that no competing interests exist.

## Introduction

*Plasmodium falciparum* is a causative agent of malaria that alternates between an insect vector and a human host. Within the human host, the parasite undergoes an asymptomatic exo-erythrocytic cycle followed by a symptomatic erythrocytic cycle. During the erythrocytic cycle, merozoite stage parasites invade red blood cells (RBCs) to initiate asexual replication. This is followed by parasite egress from cells and subsequent reinvasion of uninfected RBCs by daughter merozoites. Invasion of RBCs by parasites is a multistage process characterized by 1) initial weak engagement of the RBC by the parasite, 2) apical reorientation of the parasite and strong anchoring to the RBC, 3) tight junction formation, 4) active invasion utilizing an actin-myosin motor, and 5) shedding of the surface proteins of the parasite (*Cowman et al., 2017*). Invasion culminates in the formation of a parasitophorous vacuole surrounding the parasite within the RBC.

Erythrocyte Binding Antigen of 175 kDa (EBA-175) is a *P. falciparum* protein that binds to the host receptor glycophorin A (GpA) and this interaction has a well-defined role in anchoring the parasite during invasion of erythrocytes (*Orlandi et al., 1990*; *Orlandi et al., 1992*; *Klotz et al., 1992*; *Sim et al., 1994*; *Salinas and Tolia, 2014a*; *Liang and Sim, 1997*; *Sim, 1998*; *Duraisingh et al., 2003*; *Tolia et al., 2005*; *Wanaguru et al., 2013*; *Chen et al., 2013*; *Salinas et al., 2014b*). EBA-175 has also been shown to modify the cellular and biophysical nature of the RBC cytoskeletal complex

in preparation for efficient invasion by the parasite (*Koch et al., 2017*; *Sisquella et al., 2017*). During invasion, the tight junction formed in part by EBA-175 and GpA moves from the apical end to the posterior end of the merozoite and EBA-175 is shed from the surface of the parasite into the surrounding media during the last steps of invasion (*O'Donnell et al., 2006*).

EBA-175 is a member of the Erythrocyte Binding like (EBL) family of *Plasmodium* proteins that includes DBP, EBL-1, EBA-140, and EBA-181 (*Miller et al., 1975*; *Adams et al., 1992*; *Lobo et al., 2003*; *Maier et al., 2009*; *Batchelor et al., 2011*; *Lin et al., 2012*; *Malpede et al., 2013*; *Batchelor et al., 2014*; *Malpede and Tolia, 2014*; *Paing and Tolia, 2014*; *Salinas and Tolia, 2016*; *Chen et al., 2016*). EBL family members bind host-cell receptors via minimal binding domains comprised of either a single or double Duffy-binding like (DBL) domain (*Sim et al., 1994*; *Salinas and Tolia, 2014a*; *Liang and Sim, 1997*; *Duraisingh et al., 2003*; *Tolia et al., 2005*; *Wanaguru et al., 2013*; *Chen et al., 2013*; *Salinas et al., 2014b*; *Adams et al., 1992*; *Lobo et al., 2003*; *Maier et al., 2009*; *Batchelor et al., 2011*; *Lin et al., 2012*; *Malpede et al., 2013*; *Batchelor et al., 2014*; *Malpede and Tolia, 2014*; *Paing and Tolia, 2014*), and these domains are targets for structural vaccinology (*Chen et al., 2013*; *Batchelor et al., 2014*; *Chen et al., 2015*; *Chen et al., 2016*). The minimal binding domain of EBA-175, region II (RII), is comprised of two DBL domains, F1 and F2 (*Figure 1—figure supplement 1A*), which together bind to GpA to form a tight junction between the parasite and RBC in the initial stages of invasion (*Sim et al., 1994*; *Salinas and Tolia, 2014a*; *Tolia et al., 2005*; *Salinas et al., 2014b*). Binding of GpA induces EBA-175 dimerization to facilitate high avidity associations (*Salinas and Tolia, 2014a*; *Tolia et al., 2005*; *Wanaguru et al., 2013*; *Salinas et al., 2014b*; *Paing and Tolia, 2014*). This binding is dependent on the terminal sialic acid residues of *O*-linked glycans on the extracellular domain of GpA (*Orlandi et al., 1992*; *Sim et al., 1994*; *Salinas and Tolia, 2014a*; *Wanaguru et al., 2013*; *Salinas et al., 2014b*). Removal of sialic acid by neuraminidase or specific mutation of *O*-linked glycans ablates binding both in direct protein:protein interactions and in parasite culture (*Salinas et al., 2014b*). These specific *O*-linked glycans on GpA also allow for the specificity of EBA-175 for GpA over its highly related family member glycophorin B (*Salinas et al., 2014b*).

Merozoites use a diverse array of proteins to ensure attachment to and invasion of RBCs. Many of these proteins are involved in establishing a tight junction during invasion and antibodies to these proteins are known to prevent invasion and reduce parasite growth. The interaction with GpA can be blocked by antibodies (Abs) that target the dimer interface and glycan binding pockets of EBA-175 (*Chen et al., 2013*; *Ambroggio et al., 2013*). Ab-217, a mouse monoclonal that targets these regions, has an $IC_{50}$ of 0.1 mg/mL for parasites and can block binding of EBA-175 to GpA (*Chen et al., 2013*; *Sim et al., 2011*). Ab-218, another mouse monoclonal that targets a non-functional epitope in EBA-175, does not block this interaction and has minimal effects on parasite viability (*Chen et al., 2013*; *Sim et al., 2011*). These studies highlight the established role of membrane-anchored EBA-175 in binding GpA during *P. falciparum* invasion of erythrocytes. In addition to the EBL family, the reticulocyte-binding protein homologue (RH) family of proteins contains multiple members; each recognizing distinct receptors on RBCs during invasion. RH5, a member of the RH family, binds to the RBC receptor basigin, which is essential for invasion (*Crosnier et al., 2011*). Post-attachment, another parasite protein, AMA1, binds to a parasite protein RON2, which is inserted into the RBC plasma membrane; this interaction is crucial for the active invasion of RBCs by the parasite (*Vulliez-Le Normand et al., 2012*; *Srinivasan et al., 2011*). Both RH5 and AMA1 are promising vaccine candidates for malaria and are targets of neutralizing Abs (*Douglas et al., 2014*).

In this study, we establish a role for shed EBA-175 in RBC clustering and examine the biological consequences of this phenomenon. We embarked on this line of investigation after observing potent clustering of red blood cells upon addition of the binding domain of EBA-175 (*Koch et al., 2017*; *Salinas and Tolia, 2014a*). Here, we demonstrate that purified endogenously shed EBA-175 promotes RBC clustering and that recombinant soluble RII domain of EBA-175 is sufficient to promote RBC clustering in a GpA-dependent manner. This RII-mediated red cell clustering enhances parasite growth by providing daughter merozoites ready access to uninfected RBCs. Furthermore, we demonstrate that RII-mediated RBC clustering confers protection from neutralizing Abs that target diverse heterologous invasion proteins. This study highlights that a soluble, not a membrane-anchored, parasite protein can promote clustering of RBCs which may contribute to malaria pathogenesis by enhancing parasite growth and facilitating immune evasion. Over two hundred million new cases of malaria are reported worldwide annually with an estimated half a million deaths. The

majority of these deaths are due to severe malaria caused by *Plasmodium falciparum* parasites. EBA-175-mediated clustering may facilitate the high parasite burdens observed during severe malaria, protect parasites from antibody recognition, and prevent infected erythrocyte clearance.

## Results

### Shed EBA-175 purified from parasite culture induces RBC clustering

EBA-175 is shed by the parasite during invasion. To identify if EBA-175 has an additional role in infection after being shed post invasion, native EBA-175 was purified from a culture of Dd2 schizonts that had undergone rupture and reinvasion overnight. EBA-175 was immunoprecipitated (IP) using Ab-218 that is specific for EBA-175. Purified native EBA-175 was observed on a Coomassie stained protein gel as a band at ~175 kDa in the anti-EBA-175 lane (*Figure 1A*). The control pulldown using a non-specific antibody, Ab-8C2, did not result in a similar band pattern. Immunoblot analysis of these IP elutions with an α-EBA175 specific antibody detected a band at 175 kDa in α-EBA175 IP lane but none in control lane that (*Figure 1—figure supplement 1B*). To confirm the identity of the ~175 kDa band, mass spectrometry analysis was performed on both the control and α-EBA-175 IP elutions (*Figure 1—figure supplement 1C–D*). The most abundant peptides identified in the α-EBA175 pulldown corresponded to EBA-175 while no EBA-175 specific peptide was detected in the control pulldown (*Figure 1—figure supplement 1C–D*). Peptides identified were dispersed throughout the ectodomain of EBA-175 indicating the full-length shed protein was captured (*Figure 1—figure supplement 1D*).

The elutions from the control or the α-EBA-175 pulldown were added to RBCs in PBS for one hour at room temperature and the overall particle size of the sample analyzed by flow cytometry. The native EBA-175 induced clustering of RBCs while the control elution showed no size change (*Figure 1—figure supplement 1E*). The frequency of clusters for the native sample was noticeably higher than that of the control sample (*Figure 1—figure supplement 1E*) indicating native EBA-175 promotes clustering of RBCs. Clustering was dependent on the concentration of EBA-175 as purifications that resulted in increased yields of EBA-175 resulted in higher levels of clustering (*Figure 1—figure supplement 1E*).

### EBA-175-mediated RBC clustering is observed in parasite cultures

To assess whether endogenously shed EBA-175 could induce RBC clustering in a culture environment, NF54-GFP-Luc cultures at high parasitemia were analyzed for RBC clustering by flow cytometry analysis of particle size by FSC-A and parasite viability by GFP signal (*Figure 1D*). High parasitemia may occur in the micro environment of capillaries where infected RBCs are sequestered allowing for localized parasite multiplication. However, determining levels of native EBA-175 in blood drawn from infected individuals would be complicated by the fact that shed EBA-175 binds to uninfected RBCs and would not be freely available in serum. A large FSC-A signal was seen in the untreated NF54-GFP-Luc cultures. To assess if the large particle size seen was due to EBA-175 mediated RBC clustering, 30 µM Ab-217 orAb-218 were added to the culture after the reinvasion step to exclude the invasion-blocking effects of the Ab. Approximately 12 hr post addition of the Ab and before the next round of egress, the particle size in culture was analyzed by flow cytometry. The cultures in the presence of Ab-217 contained significantly fewer clusters in the gated population (*Figure 1E* bottom panel). All samples showed high levels of GFP demonstrating the Ab treatments had no effect on parasite invasion (*Figure 1E* top panel). This is expected as the cultures were treated with Abs at a time point to exclude invasion inhibition effects. Overall, these data suggest that a large portion of the higher size particles seen are clusters dependent on EBA-175 binding to RBCs in a GpA-dependent manner.

### Region II of EBA-175 is sufficient to induce RBC clustering

Shed EBA-175 is comprised of the entire extracellular domain of EBA-175 from regions I through VI (*Figure 1—figure supplement 1A*). Of these domains, RII contains the necessary elements to engage GpA on RBCs. To uncover the mechanism for EBA-175 induced RBC clustering, we examined if region II was sufficient to induce red cell clustering. Incubation of RBCs with picomolar and low nanomolar concentrations of RII resulted in RBC clustering by confocal live imaging of RBCs

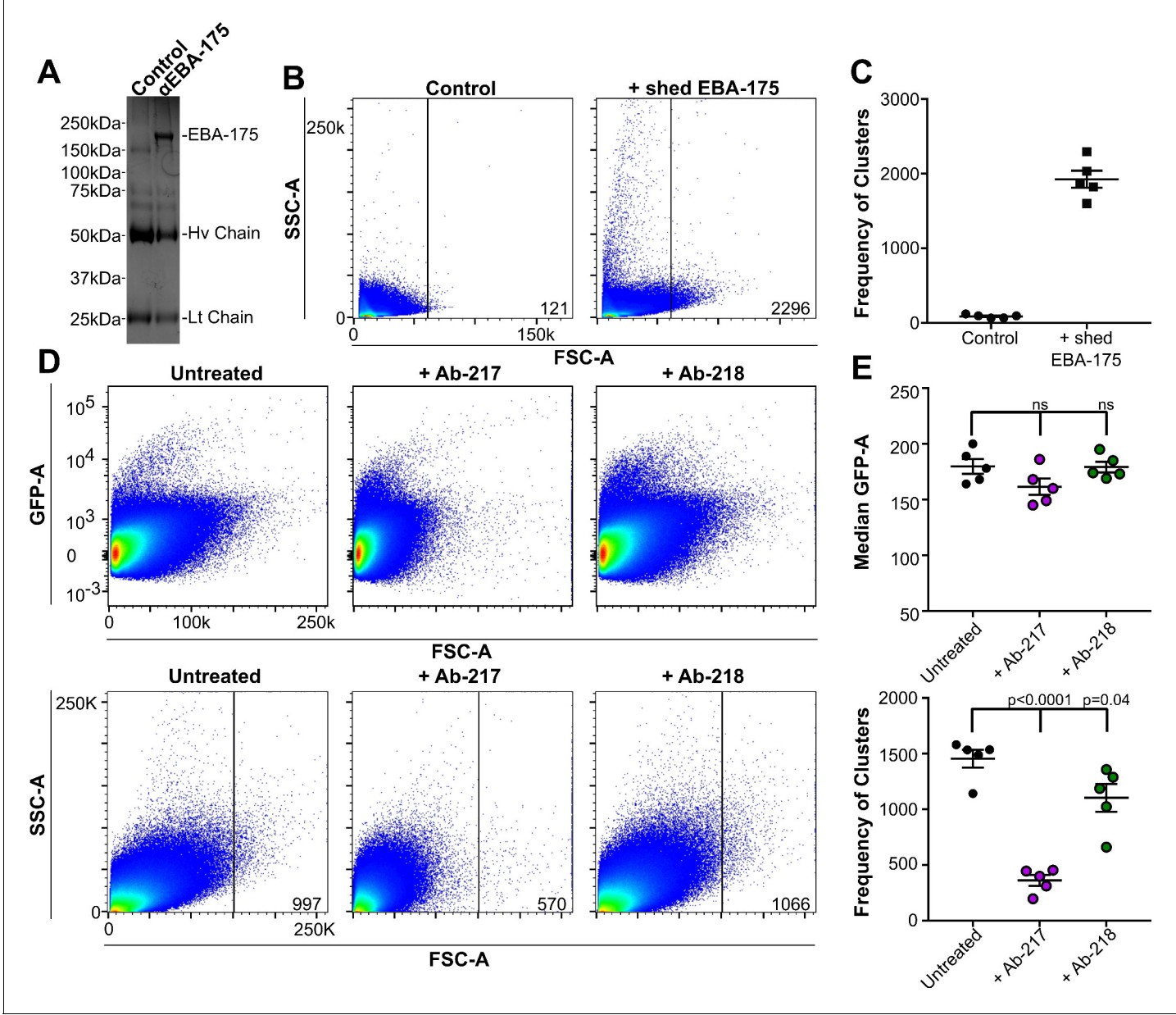

**Figure 1.** Native EBA175 promotes RBC clustering. (A-C) represent data from one biological replicate. (**A**) One representative Coomassie brilliant blue stained gel of immunopreciptations. Lane 1, control Ab Ab-8C2, lane 2, anti-EBA175 Ab-218. (**B**) Pseudocolor dot plots of the IP-elutions showing RBC clustering as observed by increasing forward scatter (FSC-A) and side scatter (SSC-A). FSC-A is correlated with the size or volume of an object while SSC-A is a measure of the internal composition of the object. Frequency of events (clusters) out of one million counts are located in bottom right corner of dot plot for one representative technical replicate. (**C**) Frequency of events (clusters) seen in the FSC-A gated population for the IP-elutions, •=Control, ■=EBA-175 treated. Values shown are for five technical replicates of one biological replicate. (Two additional biological replicates each consisting of five technical replicates can be found in *Figure 1—figure supplement 1*) (**D**) RBC clusters are observed in parasite culture as observed by FSC-A. Top panel - culture viability as shown by GFP-A for one technical replicate. Bottom panel - overall size difference of each culture as shown by FSC-A and the gated populations for one technical replicate. Frequency of events (clusters) out of one million counts are located in bottom right corner of dot plot for one representative technical replicate. (**E**) Median GFP-A signal of the total population (top) of five biological replicates each consisting of three technical replicates and frequency of events (clusters) in the gated population for five biological replicates (bottom) showing GpA dependent clustering.

DOI: https://doi.org/10.7554/eLife.43224.002

The following figure supplement is available for figure 1:

**Figure supplement 1.** Native EBA-175.

DOI: https://doi.org/10.7554/eLife.43224.003

(*Figure 2A*). This RBC clustering could be seen as low as 300 pM of RII. The picomolar and low nano-molar concentrations indicate EBA-175 RII has a potent effect in inducing RBC clustering. Furthermore, this clustering effect was observed in multiple blood types tested (*Figure 2—figure supplement 1A*)

The individual DBL domains that comprise RII, F1 and F2, were also tested for their ability to cause clustering of RBCs by flow cytometry (*Figure 2B*). Incubation of RBCs with various concentrations of RII resulted in a concentration-dependent change in particle size as characterized by an increase in forward scatter. Larger clusters were observed as higher concentration of RII (10 nM to 300 nM) were added. In contrast, F1 and F2 were unable to facilitate RBC clustering even at very high concentrations tested. This indicates that the full RII domain is necessary to facilitate clustering of RBCs consistent with prior studies that demonstrate both DBL domains are required for RBC binding (*Salinas and Tolia, 2014a*). The lack of red cell clustering in the presence of F1 or F2 also indicates that the clustering effect is specific to RII and not solely due to the addition of DBL domains to the media. Confocal live imaging of RBCs incubated with RII revealed that the increased particle size corresponded to multiple RBCs adhering to one another to form clusters (*Figure 2A & C*). Fluorescent Ab labeling showed that RII-6XHis-tag localized to the edges of the RBCs and was strongest in the junctions between adjacent RBCs (*Figure 2C*). This localization implies that RII mediates the adherence of the RBCs.

## EBA-175 RII promotes RBC clusters in a GpA dependent manner

RBC clustering was dependent on GpA on RBCs as neuraminidase treatment, an α-GpA monoclonal Ab, or Ab-217, all prevented red cell clustering (*Figure 2D* and *Figure 2—figure supplement 2A*). Neuraminidase treatment of RBCs removes the terminal sialic acid residues of GpA necessary for EBA-175 binding (*Orlandi et al., 1992*; *Salinas et al., 2014b*) and abolished RII mediated red cell clustering (*Figure 2D* panel 2), indicating that the binding of sialic acid residues in GpA by EBA-175 is a prerequisite for red cell clustering. The α-GpA Ab prevents EBA-175 access to GpA (*Salinas and Tolia, 2014a*), and also abolished RBC clustering (*Figure 2D* panel 3,4) demonstrating the requirement of EBA-175 binding to the RBC receptor. Ab-217 binds to the dimer interface and the glycan binding pocket of RII to specifically disrupt EBA-175 binding to GpA (*Chen et al., 2013*). Again, Ab-217 prevented red cell clustering (*Figure 2D* panel 5,6). Taken together, these data show that the RII mediated clustering of RBCs is dependent on GpA interacting with EBA-175.

## EBA-175 RII induced clusters are viable under physiological flow conditions

To determine if the RBC clusters observed under static conditions are present under physiological flow conditions, RBC's were incubated as before with 1, 10 or 100 nM EBA-175 RII and then subjected to 1 dyn/cm$^2$ shear stress which mimics the flow rate found in the microvasculature. Clusters were observed at all concentrations tested and the mean cluster area (1 nM - 289.7 ± 207.8; 10 nM – 223.4 ± 60.7 and 100 nM – 1135.3 ± 360.1), and mean number of clusters (1 nM – 21.3 ± 6.1; 10 nM – 88.6 ± 15.6 and 100 nM – 233.0 ± 32.4), increased in response to protein concentration (*Figure 2D*).

The pre-incubation of RBC's with EBA-175 RII generated clusters capable of withstanding 1 dyn/cm$^2$, with no clusters observed in the untreated control. As the concentration increased, the number of free RBCs was reduced, with the most striking effect observed at the highest concentration tested (100 nM), where almost all RBCs were contained within clusters, with very few remaining free (*Figure 2—figure supplement 2B*). These data generated from physiological flow assays illustrate the adhesive strength of the EBA-175 RII protein.

## Clustering of RBCs mediated by RII enhances parasite growth

We assessed whether clustering of RBCs had any effects on parasite growth in culture using diverse laboratory strains to identify strain-transcending effects. *P. falciparum* strains 3D7, Dd2, FVO/FCR1, and HB3 were incubated in the presence or absence of RII at varying concentrations from 100 pM to 10 nM of RII (*Figure 3A*) for two rounds of replication while shaking to mimic, to a limited extent, the stress and shear forces on erythrocytes in circulation. The proliferation and/or growth inhibition of the parasites was assessed using the approach of analyzing the incorporation of [$^3$H]-hypoxanthine

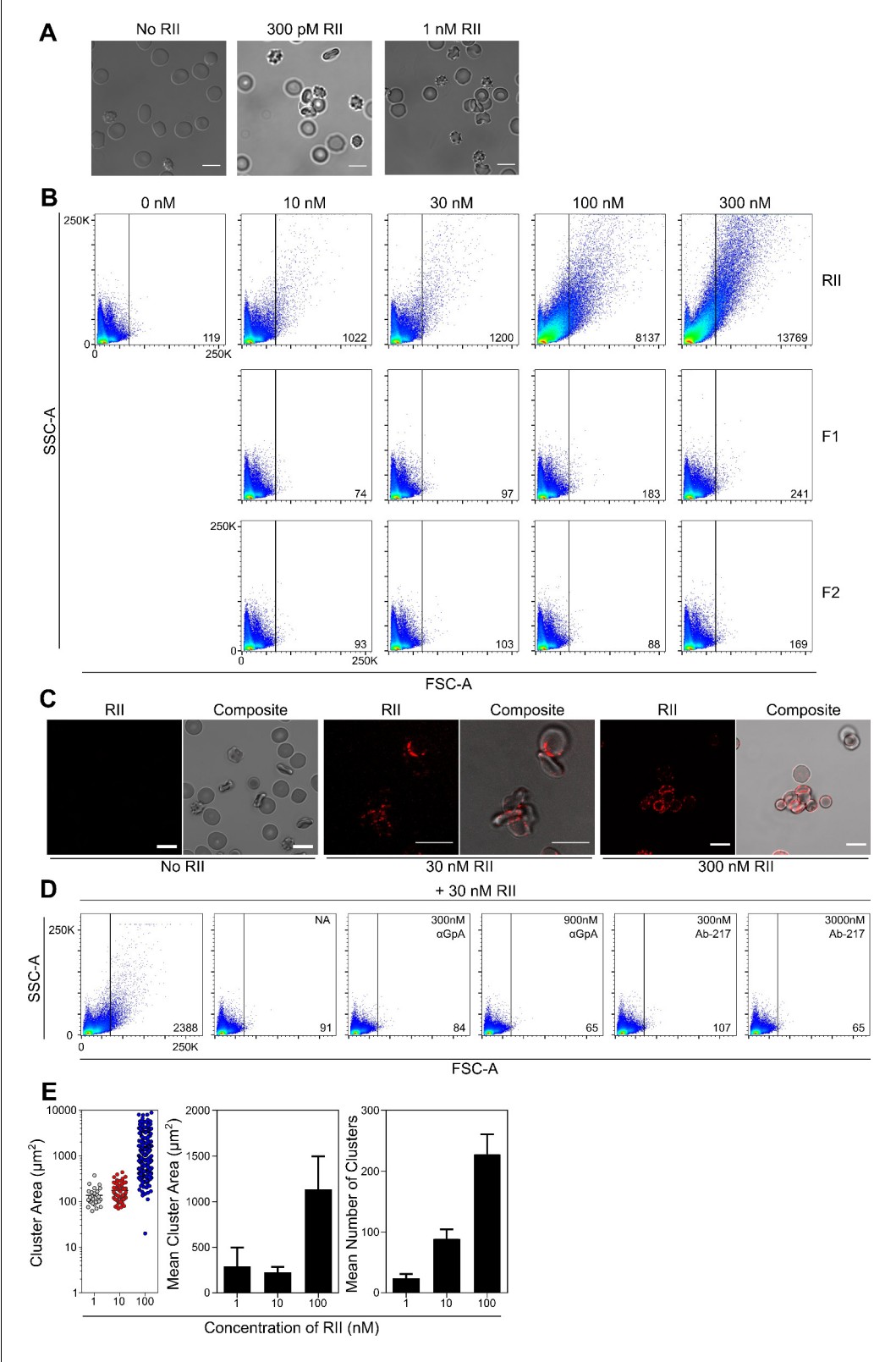

**Figure 2.** EBA-175 RII binds and promotes RBCs to form clusters through interactions with GpA. (**A**) Clustered RBCs adhere to one another via RII at 300pM and 1 nM concentrations. Scale bars are 10 μm. (**B**) RII, but not the individual DBL domains F1 or F2, induces RBC clustering in a concentration dependent manner as observed by FSC-A. Frequency of events (clusters) out of 100,000 counts are located in bottom right corner of dot plot for each sample when gated according to the no protein control (top left). One representative biological replicate out of three is presented. (**C**) Clustered RBCs

*Figure 2 continued on next page*

*Figure 2 continued*

adhere to one another via RII at the interface between each red cell. Scale bars are 10 μm. (D) Inhibition of particle size shift of RBCs incubated with RII by neuraminidase (NA, panel 2), anti-GpA Ab (panel 3 and 4), and anti-RII Ab-217 as observed by FSC-A (panel 5 and 6). Frequency of events (clusters) out of 100,000 counts are located in bottom right corner of dot plot for each sample when gated according to the no protein control (top left). One representative biological replicate out of three is presented. (E) Analysis of cluster size for varying concentrations of EBA-175 RII under flow conditions of 1 dyn/cm$^2$. Left hand panel shows representative data from one experiment. Each circle represents a single cluster area (μm$^2$). Middle panel shows mean cluster size for each concentration, and right hand panel shows mean number of clusters per concentration for three independent experiments ± S.D.

DOI: https://doi.org/10.7554/eLife.43224.004

The following figure supplements are available for figure 2:

**Figure supplement 1.** EBA-175-RII mediates similar levels of parasite growth in O, A, and B blood types.

DOI: https://doi.org/10.7554/eLife.43224.005

**Figure supplement 2.** RII binds RBCs and forms stable clusters under flow conditions.

DOI: https://doi.org/10.7554/eLife.43224.006

into parasite nucleic acids after two rounds of replication to get a better signal-to-noise ratio. A significant increase in parasite growth was observed when RII was present at concentrations as low as 300 pM (p<0.05, in Dd2 strain) and could be seen up to 10 nM (*Figure 3A*). The significant growth enhancement at 1 nM, p<0.001, was also observed in a NF54 line expressing a firefly luciferase transgene, NF54-GFP-luc (*Figure 3A* panel 5), demonstrating that the growth enhancement phenotype was not due to non-specific effects of RII on [$^3$H]-hypoxanthine incorporation into parasite nucleic acids. Further, the growth enhancement effect was observed in multiple blood types tested (*Figure 2—figure supplement 1B*).

We examined if the individual DBL domains of RII could also facilitate enhanced growth through RBC clustering. An expanded protein concentration range from 10 nM to 1 μM of RII, F1, and F2 was used to observe any potential effects. A significant increase in parasite growth was observed when RII was present at concentrations from 10 nM to 100 nM with an optimal concentration of 30 nM (*Figure 3B*). Assessment of F1 and F2 revealed that F1 was unable to promote growth, while F2 evidenced moderate growth promoting activity at concentrations higher than 100 nM (*Figure 3B* and *Figure 3—figure supplement 1*) consistent with high avidity binding of EBA-175 to GpA requiring both DBL domains (*Salinas and Tolia, 2014a*; *Tolia et al., 2005*; *Wanaguru et al., 2013*). Therefore, the observed growth enhancement in parasite culture requires an intact RII with both DBL domains.

## Growth enhancement is dependent on RII binding to GpA

To analyze whether growth enhancement was due to RBC clustering via the specific interaction between EBA-175 and GpA, Ab-217 or a control Ab (Ab-DBP) were added to the parasite cultures. Ab-217 blocked the RII-mediated growth enhancement (*Figure 3C*, *Figure 3—figure supplement 2*, and *Figure 2—figure supplement 1C*) while the Ab-DBP had no effect on growth. The concentrations of Ab-217 used were far below the IC$_{50}$ on invasion (*Sim et al., 2011*) and did not cause any effects on invasion. Taken together, the data suggest that EBA-175 promotes RBC clustering in a GpA-dependent manner which results in a higher growth of parasite culture, due to the increase in invasion into adjacent adherent uninfected RBCs.

## Visualization of EBA-175 RII mediated RBC clustering in parasite culture

Microscopy analysis of NF54-GFP-luc cultures in the presence of nanomolar RII revealed large clusters of uninfected RBCs associated with infected RBCs (*Figure 3D*). More than one parasite was observed in the clusters and these tend to group together in adjacent RBCs. This suggests that multiple daughter merozoites, arising from a single schizont, invade adjacent uninfected RBCs within a cluster. Therefore, red cell clustering may provide daughter merozoites ready access to uninfected RBCs by bringing the uninfected RBCs in close proximity to egressing merozoites.

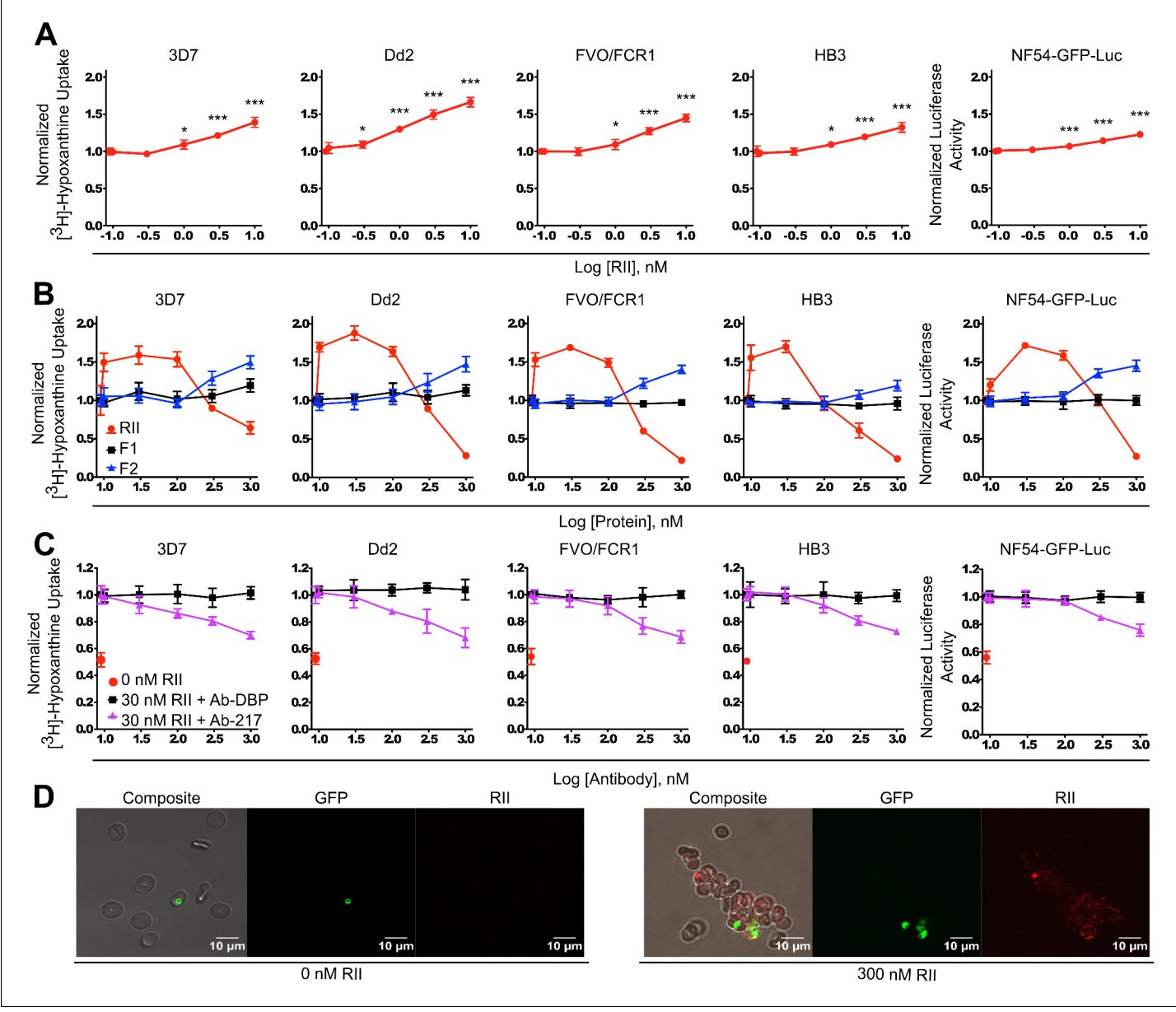

**Figure 3.** Recombinant EBA-175 RII causes RBC clustering in parasite culture and enhances parasite growth. (**A**) Effects of picomolar to low nanomolar RII on parasite growth assessed by [³H]-hypoxanthine uptake for 3D7, Dd2, FVO/FCR1, HB3 strains, and the luciferase activity for NF54-GFP-luc strain (far right). Data shown are mean ± SD of three biological replicates for all strains. Significance (*, p<0.05; ***, p<0.001) (**B**) Uptake of [³H]-hypoxanthine into parasite nucleic acids by 3D7, Dd2, FVO/FCR1, HB3 strains, and the luciferase activity in NF54-GFP-luc strain (far right), in the presence or absence of RII, F1 or F2 at varying concentrations (10 nM – 1 µM range) for 96 hr. Results are shown as mean ± SD of three biological replicates. (**C**) Ab-217 (purple line) inhibition of RII-mediated growth enhancement, control Ab Ab-DBP (black line). Results are shown as mean ± SD of three biological replicates. (**D**) Microscopy analysis of NF54-GFP parasites (green) in RII-induced clusters (red).

DOI: https://doi.org/10.7554/eLife.43224.007

The following figure supplements are available for figure 3:

**Figure supplement 1.** RII enhances parasite growth.
DOI: https://doi.org/10.7554/eLife.43224.008
**Figure supplement 2.** Parasite growth enhancement is due to the specific interaction of EBA-175 with GpA.
DOI: https://doi.org/10.7554/eLife.43224.009

## EBA-175-RII-mediated RBC clustering confers parasites protection from diverse neutralizing abs

We asked whether RBC clustering confers parasites a survival advantage, other than growth enhancement, in the presence of known neutralizing Abs during subsequent rounds of invasion following cluster formation. We assessed the effect of RII-mediated red cell clustering on parasite viability by α-AMA1 neutralizing Abs (N3-2D9 and N4-1F6), and the α-RH5 neutralizing Ab 9AD4 (36). The two α-AMA1 antibodies blocked the growth of 3D7 parasites by 20% and 50%, respectively in the absence of RII. Strikingly, addition of 30 nM RII strongly reduced the ability of these Abs to neutralize parasites (*Figure 4A*). The FVO-FCR1 parasites are more sensitive to AMA-1 Abs with 25% and 65% inhibition by N3-2D9 and N4-1F6, respectively (*Figure 4B*). As in 3D7, this inhibitory effect of AMA-1 neutralizing Abs on FVO-FCR1 was also significantly mitigated in the presence of 30 nM RII (*Figure 4B*). The protection from neutralizing Abs was not due to artifacts of the hypoxanthine

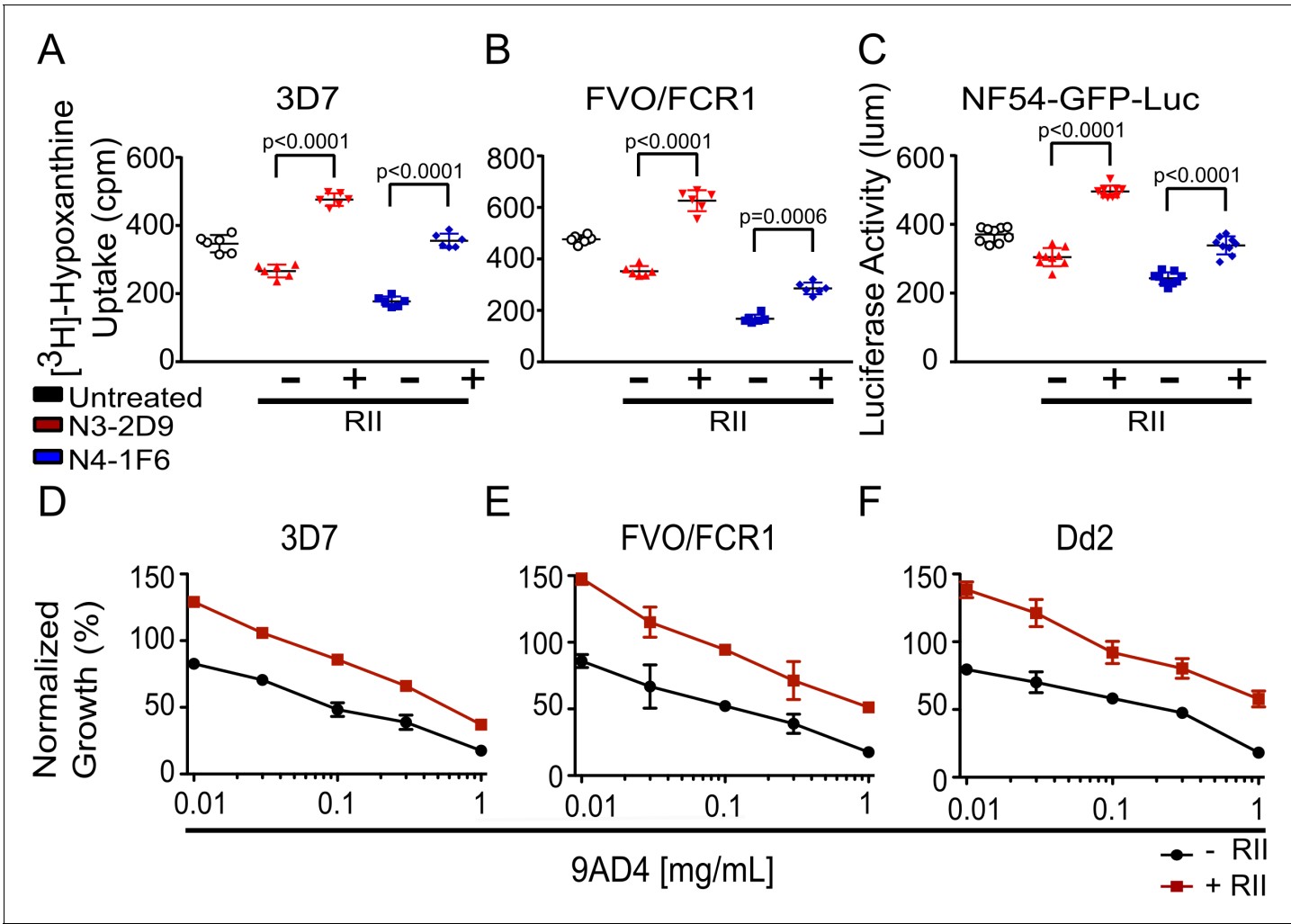

**Figure 4.** EBA-175 RII-mediated RBC clustering provides parasites a growth advantage and confers protection from anti-AMA1 and anti-RH5 neutralizing antibodies. (A–C) The presence of 30 nM RII in (A) 3D7, (B) FVO/FCR1, and (C) NF54-GFP-luc cultures mitigated the inhibitory effects of the anti-AMA1 neutralizing antibodies N3-2D9 and N4-1F6 at 10 mg/mL. (D–F) A 30 nM RII mitigated the ability of anti-RH5 neutralizing Ab, 9AD4, to inhibit growth of (D) 3D7, (E) FVO/FCR1 and (F) Dd2 cultures at various concentrations. Results are shown as mean ± SD of three biological replicates and significance was determined as described in Materials and methods.

DOI: https://doi.org/10.7554/eLife.43224.010

The following figure supplement is available for figure 4:

**Figure supplement 1.** EBA-175-mediated RBC clustering confers parasites a survival advantage by preventing neutralization by antibody 9AD4.
DOI: https://doi.org/10.7554/eLife.43224.011

incorporation assay as a similar elimination of Ab neutralization was observed in NF54-GFP-luc parasites measured using luciferase activity (*Figure 4C*).

We next assessed a recombinant α-RH5 mouse Ab that includes the variable region of Ab 9AD4 which potently blocks RH5 engagement of the receptor basigin (*Douglas et al., 2014*). In our assays, 9AD4 exhibited dose-dependent inhibition with maximal inhibition of 80% observed at 1 mg/mL of Ab concentration for 3D7, FVO/FCR1, and Dd2 in the absence of RII (*Figure 4D–F*). Strikingly, and similar to the results seen with the α-AMA1 Abs, 30 nM RII was able to protect against inhibition by 9AD4 in the parasite lines 3D7, FVO/FCR1, and Dd2 at all concentrations tested (*Figure 4D–F* and *Figure 4—figure supplement 1*). These results demonstrate that RII-mediated RBC clustering can protect parasites from neutralizing Abs that target diverse heterologous antigens, providing a mechanism for immune evasion.

## Discussion

This study uncovered a biological consequence of shed EBA-175 in red cell clustering that leads to enhanced growth and confers parasite protection from neutralizing Abs. Immune evasion has long been suspected in *Plasmodium* infections. However, neither the molecular basis for this process nor how it might relate to disease pathogenesis have been fully defined. The results support a model (*Figure 5*) in which shed EBA-175, perhaps in conjunction with other membrane-anchored proteins, facilitates formation of RBC clusters to benefit parasite growth and survival. During infection, EBA-175 on the surface of parasites interacts with GpA on the surface of RBCs during invasion (*Sim et al., 1994*). Post invasion, we and others (*O'Donnell et al., 2006*) have shown that EBA-175 is released into the surrounding environment where it binds to uninfected RBCs, facilitating EBA-175- and GpA-dependent clustering of naïve RBCs around the infected RBCs. Red cell clustering subsequently confers protection from antibodies and the immune system, promoting efficient invasion of adjacent uninfected RBCs by daughter merozoites. The enhancement of invasion into adjacent uninfected RBCs leads to an increase in parasite growth and the process perpetuates through the next round of uninfected RBC recruitment. We observed red cell clustering and growth enhancement at picomolar concentrations of EBA-175 implying that this phenomenon could exist in vivo. These concentrations are likely achievable during malaria infection particularly in the capillaries and extremities of blood vessels. This may be a mechanism, along with rosetting afforded by membrane-anchored proteins, for the high parasitemia seen in severe malaria through enhancement of parasite growth and immune evasion.

This mechanism is also direct evidence for active immune evasion by *Plasmodium* parasites and offers new insights into parasite-host interactions and vaccine development. This phenomenon may be relevant for the interpretation of the data generated from EBA-175-based vaccines in clinical trials. Additional studies are needed to examine whether this phenomenon extends to Abs that target additional parasite proteins, and if antibodies that occur in natural infection block EBA-175 or other EBL family member-mediated red cell clustering. The model proposed is consistent with synergistic inhibition (*Ord et al., 2012*; *Williams et al., 2012*) observed when α-EBA-175 and α-RH5 antibodies are used in combination. The model indicates that the synergistic inhibition arises as EBA-175 red cell clustering is prevented, allowing for increased inhibition by α-RH5 Abs. Additionally, other EBL family members, along with numerous other parasite proteins that are vaccine candidates, are released during infection. The model and data presented here forms a strong foundation for the future study of clustering and its role in immune evasion in patient populations. In addition, this study forms a framework to examine the breadth and depth of proteins that can facilitate RBC clustering and immune evasion.

The clustering phenomenon reported here exhibits similarities to *P. falciparum* rosetting attributed to the parasite membrane proteins PfEMP1 (*Rowe et al., 1997*; *Moll et al., 2015*; *Chen et al., 1998*), RIFINs (*Goel et al., 2015*) and STEVORs (*Niang et al., 2014*). PfEMP-1 is related to the EBL family as it contains multiple DBL domains including the DBLα domain which binds to the RBC receptor CR1 during sequestration of rosettes (*Rowe et al., 1997*). RIFINs bind to blood group A RBCs to form rosettes (*Goel et al., 2015*) and may bind to GpA on RBCs to mediate rosetting (*Goel et al., 2015*). STEVORs bind to RBCs through glycophorin C (*Niang et al., 2014*). Similarly, the EBL protein EBA-140 also binds glycophorin C and may serve a parallel role (*Lobo et al., 2003*; *Malpede et al., 2013*). However, evidence of growth enhancement and immune protection from

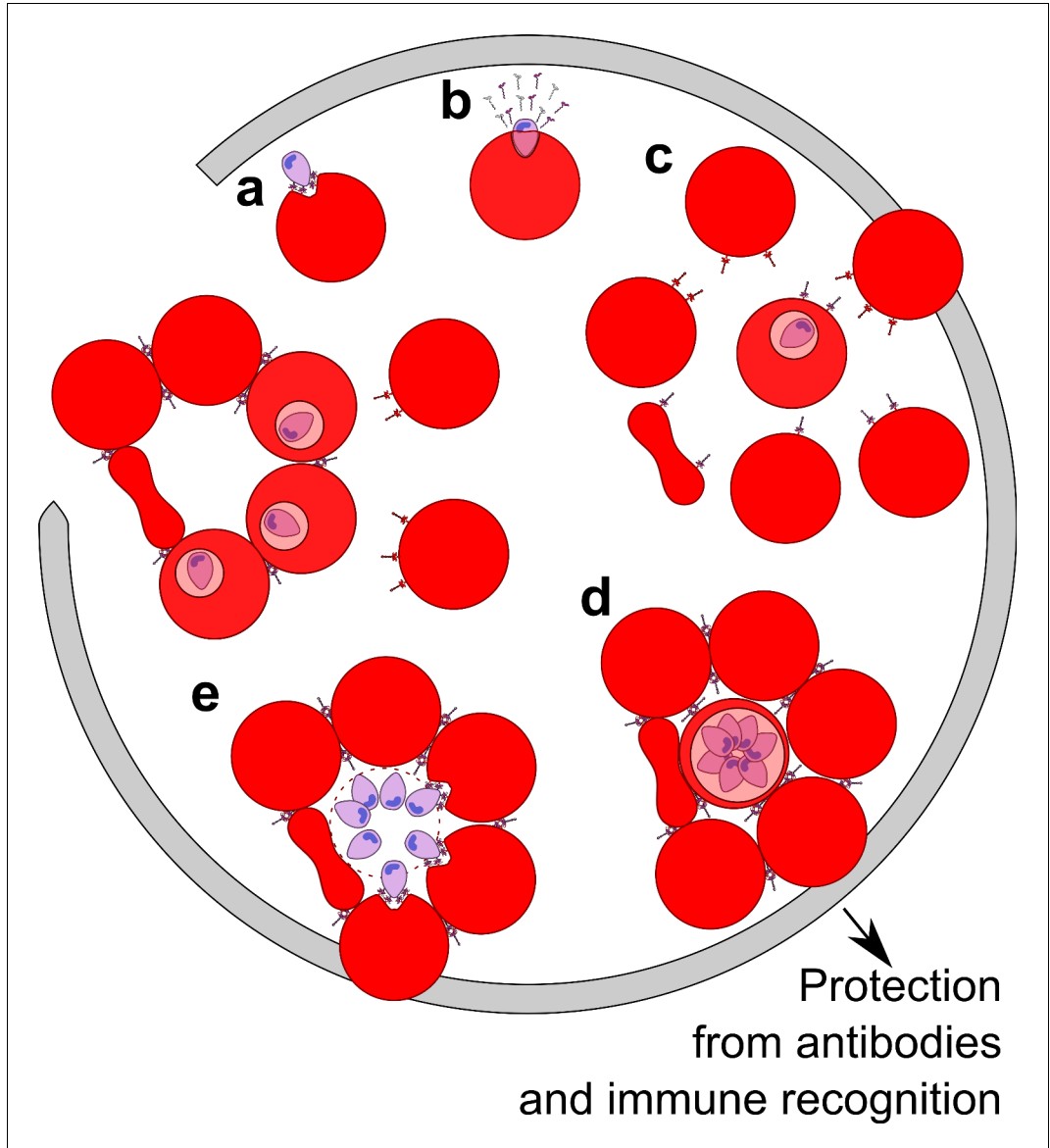

**Figure 5.** Model of the parasite survival and immune evasion mediated by EBA-175. (**A**) Merozoite (purple) secretes EBA-175 to the apical end for engagement of GpA. (**B**) Post invasion, EBA-175 is cleaved from the surface and shed. (**C**) Shed EBA-175 diffuses to neighboring RBCs and engages GpA resulting in RBC recruitment and red cell clustering (**D**). (**E**) RBC clustering enables daughter merozoites easy access to an uninfected RBC for the next round of invasion. Gray arrow indicates progression through one round of invasion and replication.
DOI: https://doi.org/10.7554/eLife.43224.012

Abs due to rosetting has yet to be reported. Further studies are necessary to determine if additional members of the EBL family can also promote RBC clustering and enhance growth of parasites. It is also important to note that both PfEMP1 and RIFINs appear to prefer type A blood for rosetting and show minimal binding to type O blood (*Goel et al., 2015*; *Vigan-Womas et al., 2012*). Red cell clustering defined here is independent of blood type and provides a novel pathway for RBC clustering that complements PfEMP1, RIFINs or STEVORs.

Individuals in regions endemic for malaria develop partial immunity to all parasite invasion ligands and yet incidences of severe malaria still occur. The evidence we provide here offers three mechanisms by which this may occur. The first mechanism is that EBA-175-mediated enhancement of growth simply outpaces the ability of the immune system to target parasites and prevent invasion. The second is that RBC clustering limits the time window of antigen exposure to neutralizing

antibodies during the rapid invasion of an adjacent RBC, preventing Ab access. The third is that RBC clustering may block physical access of the antibodies to parasites during invasion. These mechanisms are not mutually exclusive and the true mechanism may be a combination of the three. These studies underline the need to explore the role of the EBL family of proteins in pathogenesis of malaria beyond the tight junction formation.

## Materials and methods

### Protein expression and purification

RII, RII-6xHis, F1, and F2 were expressed and purified as described (*Salinas and Tolia, 2014a*; *Chen et al., 2013*). Briefly, recombinant proteins were expressed as insoluble proteins in *E. coli* and refolded. Refolded proteins were purified by ion exchange chromatography and gel filtration in PBS buffer.

### Antibody purification and generation

Antibodies Ab-217 and Ab-218 (anti-EBA-175-RII), N3 and N4 (anti-AMA1), and 8C2 were purified from hybridoma supernatants using Protein A agarose resin (Gold Biotechnology, St. Louis, MO). Antibodies were buffer exchanged into PBS after purification and sterile filtered for use in parasite cultures. The heavy and light chain variable segments of 9AD4 as deposited in PDB 4U0R (*Wright et al., 2014*) were grafted onto a mouse IgG2a backbone (AF466698.1) and a mouse kappa light chain backbone (AM745100.1), respectively, codon-optimized and cloned into mammalian expression vector pHLSec. A 1:1 ratio of 9AD4 heavy and light chain constructs were transiently transfected into 293 F cells using polyethylenimine (PEI) at a ratio of 1:3 DNA:PEI. Five days post transfection, cultures were harvested and the 9AD4 IgG was purified from the supernatant by Q sepharose resin (GE Healthcare, Marlborough, MA) followed by protein A resin. The purified protein was then buffer exchanged into PBS and sterile filtered for use in parasite cultures. The control Ab-DBP (α-PvDBP) Ab was generously provided by the lab of John Adams (University of South Florida).

### Flow cytometry of native proteins

RBCs at 2% hematocrit in PBS were utilized in the flow cytometry assays. Elutions from the two immunoprecipitation reactions were added to RBCs for 1 hr at room temperature. The samples were analyzed by a BD FACSCanto (BD Biosciences, San Jose, CA, USA) machine with one million events recorded for each technical replicate. Each biological replicate was read five times for a total of five technical replicates. Three biological replicates each consisting of five technical replicates were analyzed by FlowJo and gated according to the no protein control.

### Flow cytometry of parasite cultures

NF54-GFP-luc parasites were grown without media change for 5 days. Antibodies Ab-217 and Ab-218 were added to cultures 12 hr prior to analyzing with a BD FACSCanto (BD Biosciences, San Jose, CA, USA), prior to the next round of egress. One million events were recorded for each technical replicate. Five biological replicates each consisting of three technical replicates were collected. All data was analyzed by FlowJo and gated according to the no protein control.

### Flow cytometry of recombinant proteins

Red blood cells at 2% hematocrit in PBS were utilized in the flow cytometry assays. RII, F1, or F2 were added at varying concentrations to the RBCs for 30 min at room temperature and then washed three times with PBS. The samples were then run using a BD FACSCanto (BD Biosciences, San Jose, CA, USA) and analyzed with FlowJo software. One hundred thousand events were recorded for each biological replicate with three biological replicates total. Antibodies at varying concentrations were preincubated with RII at room temperature for 30 min before incubation with RBCs before analysis. For experiments with neuraminidase, 50:50 (v/v) RBCs in RPMI were treated with 10 mU/mL neuraminidase (Sigma, St. Louis, MO) for 1 hr at 37°C. All data was analyzed by FlowJo and gated according to the no protein control. For those experiments with O, A, and B blood types the same procedure was followed as described above with one million counts read per sample.

## Cluster formation assays under flow

The cluster formation assays under flow were carried out as described previously (*Adams and Rowe, 2013*; *Lennartz et al., 2015*) with minor modifications. Briefly, a µ-slide VI 0.4 (dimensions: 400 µm × 3.8 mm×17 mm) constructed of optical quality polymer. The flow rate used generates a wall shear stress of 1 dyn/cm$^2$, which mimics the wall shear stress in the microvasculature (*Yipp et al., 2000*). To generate the flow rate, chips were connected to an NE-1002X syringe pump (World Precision Instruments, U.K.) All measurements of clusters under physiological shear stress where conducted within the area of homogenous shear stress. For µ-slide I 0.4, this is at least 400 µm away from the channel walls; following manufacturer's recommendations, all observations are made down the central line of the slide (see Application Note AN-11, https://ibidi.com/). The µ-slide VI chambers (ibidi, Germany) were blocked overnight at 4°C with PBS/1% BSA. The following day, 5 ml 1% Ht 0 + RBC in RPMI (pH7.2–7.4) was prepared. The suspension was added to the reservoir syringe and allowed to flow over the µ-slide VI for 5 min at 1 dyn/cm$^2$, before stopping the flow and immediately capturing both bright-field images from 10 consecutive fields with a 20 × objective, using a Leica DM1 inverted light microscope. In each experiment, the RBC without protein sample was tested first, followed by the samples containing EBA-175 RII protein starting from the lowest concentration (1 nM) followed by 10 nM then 100 nM.

## Quantification of cluster under flow

Using ImageJ software (http://rsb.info.nih.gov/ij/), each of the 10 images corresponding to 10 fields per sample were analysed, and the size of clusters was measured. In preparation for measurement, the threshold settings were adjusted to highlight the clusters, in order to allow the margins of the cell clusters to be identified more clearly. Briefly, the captured images were processed using brightness and contrast, followed by threshold to enhance the visualization of clusters against the background. Clusters were selected using the Analyze/Measurement settings, with the 'wand' tool, which highlights the circumference of the selected rosette and calculates the area (µm$^2$). All of the measurements from each image were added and the results expressed as mean cluster area (µm$^2$) and mean number of clusters from 10 fields, from three independent experiments.

## Statistical analysis and graphing

Data were analyzed and graphs prepared using GraphPad Prism v7.0. (GraphPad Software Inc.). Mean cluster values from at least three independent experiments for each concentration were compared by one-way ANOVA, with $p < 0.05$ being taken as statistically significant.

## Parasite cultures

The *P. falciparum* strains 3D7, Dd2, FVO/FCR1, and HB3 were cultured in fresh human O$^+$red blood cells (Valley Biomedical, Barnes Jewish and St. Louis Children's Hospital Blood Banks) at 2% hematocrit in rich medium (RPMI 1640 supplemented with 27 mM NaHCO$_3$, 11 mM glucose, 0.37 mM hypoxanthine, 10 µg/mL gentamicin, 5 g/L Albumax). For assessing blood type influence on RII-mediated growth, parasites were cultured in fresh human A$^+$, B$^+$, or O$^+$red blood cells at 2% hematocrit in rich medium. Synchrony of parasites was maintained with successive rounds of 5% sorbitol treatment as described previously (*Drew et al., 2008*). A transgenic *P. falciparum* NF54 parasite line expressing a GFP-firefly luciferase transgene (NF54HT-GFP-luc - BEI/ATCC, Manassas, VA) was cultured in rich medium with 10 nM WR99210 (Jacobus Pharmaceutical Co, Princeton, NJ). For assessment of natively shed EBA-175, NF54HT-GFP parasites were cultured without media change for 5 days to allow accumulation of shed EBA-175 in the growth media.

## Parasite growth assay using [$^3$H]-hypoxanthine

Semi-synchronous 3D7, Dd2, FVO/FCR1, and HB3 in trophozoite/schizont stages at 2% hematocrit were plated in 96-well plates with a starting parasitemia of 2.5–3.0% and the cultures were kept shaken throughout the experiment to mimic in vivo conditions and limit the number of multiple invasion events into a single RBC. Purified RII and/or antibodies were immediately added to cultures after plating. These cultures were allowed to progress through 1.5 cycle (72 hr) in assay medium (90% hypoxanthine-free rich medium mixed with 10% rich medium). [$^3$H]-hypoxanthine (0.5 µCi/well) was then added and parasites were incubated for a further 18–24 hr in assay medium. The assay

duration (96 hr) corresponds to approximately two intraerythrocytic developmental cycles. The assay is stopped by harvesting the well contents onto a filter paper (Skatron FilterMAT 11731) using a semiautomatic cell harvester (Skatron Instruments, Sterling, VA). After air dried, the mats were analyzed using Beckman LS6000 IC liquid scintillation counter. Each [³H]-hypoxanthine incorporation measurement was performed in duplicate (in accordance with ALARA for the use and disposal of radioactive waste) for three independent times.

## Parasite growth assay using luciferase substrate

To analyze growth by luciferase activity, semi-synchronous NF54HT-GFP-luc parasites at 1% parasitemia and 2% hematocrit were seeded in a 96-well plate and cultured in rich medium with 10 nM WR99210. After ~96 hr of incubation in continuous presence or absence of RII and antibodies, 25 µL of well contents was mixed with 25 µL of Bright-Glo Luciferase Assay substrate (Promega, Madison, WI). The mixture was thoroughly mixed for 10 min at 37°C using an Eppendorf MixMate and luminescence was measured using Cytation3 Cell Imaging Reader (BioTek Instruments, Winooski, VT). Each measurement was performed in triplicate for three independent times and data were analyzed and plotted using Prizm 7.0 software.

## Purification of endogenously shed EBA-175

Upon rupture of Dd2 schizonts and reinvasion overnight, the culture medium was harvested, concentrated 40x by centrifugation using 10 kDa molecular mass cutoff concentrators (MilliporeSigma). A new batch of RBCs were then added and mixed for 30 min at room temperature to promote EBA-175 binding. The RBCs were collected and incubated in 2 volumes of RPMI-1640 supplemented with 1.5 M NaCl to elute all surface bound protein. The eluant was collected and NaCl concentration was adjusted back to 150 mM with PBS. EBA-175 was immunoprecipitated with Ab-218 or, as a control, Ab-8C2, crosslinked to protein A agarose beads using Pierce Crosslink IP kit. Protein A beads-immune complexes were recovered, washed with TBS, and bound proteins were eluted with gentle elution buffer (Pierce). The elution buffer was buffer exchanged to PBS for biological assays.

## Mass spectrometry analysis of purified endogenous EBA-175

Eluted IP samples were separated on 4–12% gradient SDS gel electrophoresis and the protein bands were visualized with Coomassie Brilliant Blue stain. Bands appearing between 150–200 kDa from the 8C2 pulldown or 218 pulldown lane were excised and sent for analysis at the FingerPrints Proteomics Facility at University of Dundee. Briefly, the gel slices were subjected to overnight (16 hr) trypsin digestion (Modified Sequencing Grade, Roche). Peptides were then extracted from gel and dried in SpeedVac (Thermo Scientific), resuspended in 50 µL 1% formic acid and 2.5% Acetonitrile, centrifuged and transferred to a HPLC vial. Fifteen microliters of sample was injected into Ultimate 3000 RSLCnano system (Thermo Scientific) coupled to a LTQ OrbiTrap Velos Pro (Thermo Scientific). Peptides were initially trapped on an Acclaim PepMap 100 (C18, 100 µM x 2 cm) and then separated on an Easy-Spray PepMap RSLC C18 column (75 µM x 50 cm) (Thermo Scientific). Sample was transferred to mass spectrometer via an Easy-Spray source with temperature set at 50 °C and a source voltage of 1.9 kV. Orbitrap XL.RAW files converted to MSF files (Proteome Discoverer Version 2.2) and the extracted data were searched against PlasmoDB using Mascot Search Engine (Version 2.3.2).

## Microscopy

Red blood cells at 2% hematocrit in PBS were utilized for all microscopy studies. RBCs alone were incubated with various concentrations of RII for 1 hr at room temperature before imaging. To detect the localization of RII, RII-6xHis at 30 nM or 300 nM was added to the RBCs and incubated for 30 min at 37°C followed by three washes with PBS. Anti-6xHis Ab (Invitrogen) was added at a dilution of 1:100, incubated for 30 min at 37°C, and the samples washed three times with PBS. Anti-mouse IgG conjugated to Alexafluor 546 (Invitrogen) was added at a dilution of 1:200, incubated for 30 min at 37°C, and the samples washed three times with PBS. The samples were then resuspended in PBS, diluted 1:100 in PBS and transferred to a 96 well glass bottom plate. Samples were imaged at 63x magnification using Zeiss LSM880 laser scanning confocal microscope.

To visualize RBC clusters in NF54HT-GFP-Luc parasite culture, RII-6xHis was added 48 hr prior to imaging. Just before imaging, 1:100 diltuion of anti-6xHis Ab (Invitrogen) was added to cultures and incubated for 30 min at 37°C. Anti-mouse IgG conjugated to Alexafluor 546 (Invitrogen) was added to the culture for a final concentration of 1:200 and incubated for 30 min at 37°C. The cultures were then washed two times with PBS and resuspended in 100 μL of PBS. The samples were then diluted 1:100 in PBS and transferred to a 96 well glass bottom plate. Samples were imaged as described above.

### Statistical analysis

Data were tested for normality by one, or a combination of the Kolmogorov-Smirnov test, Shapiro-Wilk test, and D'Agostino-Pearson test. All data were found to be normally distributed and significance was determined by one-way ANOVA with Dunnett's multiple comparison tests.

## Acknowledgements

This work was supported by the Intramural Research Program of the National Institute of Allergy and Infectious Diseases, by the Extramural Research Program of the National Institute of Allergy and Infectious Diseases grant AI080792 and the Burroughs Wellcome Fund to NHT. Antibodies used in the studies were produced at the Facility of the Rheumatic Diseases Core Center at Washington University School of Medicine, funded by the National Institute of Arthritis and Musculoskeletal and Skin Diseases (Award Number P30AR048335). Imaging studies were performed at the Molecular Microbiology Imaging Facility at the Washington University School of Medicine. The mass spectrometry analysis was performed by the FingerPrints Proteomics Facility at the University of Dundee which is supported by the Wellcome Trust Technology Platform Award (097945/B/11/Z). We thank J Patrick Gorres for his assistance in preparing this manuscript for publication.

## Additional information

### Funding

| Funder | Grant reference number | Author |
| --- | --- | --- |
| Burroughs Wellcome Fund | | Niraj H Tolia |
| National Institute of Allergy and Infectious Diseases | Intramural Research Program | Niraj H Tolia |
| National Institute of Allergy and Infectious Diseases | Extramural Research Program AI080792 | Niraj H Tolia |

The funders had no role in study design, data collection and interpretation, or the decision to submit the work for publication.

### Author contributions

May M Paing, Data curation, Formal analysis, Validation, Investigation, Methodology, Writing—original draft, Writing—review and editing; Nichole D Salinas, Data curation, Formal analysis, Validation, Investigation, Methodology, Writing—original draft, Project administration, Writing—review and editing; Yvonne Adams, Data curation, Formal analysis, Investigation, Methodology, Writing—review and editing; Anna Oksman, Investigation, Methodology; Anja TR Jensen, Supervision, Methodology, Writing—review and editing; Daniel E Goldberg, Supervision, Methodology, Project administration, Writing—review and editing; Niraj H Tolia, Conceptualization, Resources, Supervision, Funding acquisition, Investigation, Methodology, Writing—original draft, Project administration, Writing—review and editing

### Author ORCIDs

Anja TR Jensen ⓘ http://orcid.org/0000-0002-4004-7554
Daniel E Goldberg ⓘ http://orcid.org/0000-0003-3529-8399
Niraj H Tolia ⓘ http://orcid.org/0000-0002-2689-1337

**Decision letter and Author response**
Decision letter https://doi.org/10.7554/eLife.43224.015
Author response https://doi.org/10.7554/eLife.43224.016

## Additional files

### Supplementary files
• Transparent reporting form
DOI: https://doi.org/10.7554/eLife.43224.013

### Data availability
All data generated or analysed during this study are included in the manuscript and supporting files.

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
