## [Decision Letter]

[Editors’ note: a previous version of this study was rejected after peer review, but the authors submitted for reconsideration. The first decision letter after peer review is shown below.]

Thank you for submitting your work entitled "Shed EBA-175 mediates red blood cell clustering that enhances malaria parasite growth and enables immune evasion" for consideration by *eLife*. Your article has been reviewed by three peer reviewers, one of whom is a member of our Board of Reviewing Editors, and the evaluation has been overseen by a Senior Editor. The following individuals involved in review of your submission have agreed to reveal their identity: Richard Culleton (Reviewer #2).

Our decision has been reached after consultation between the reviewers. Based on these discussions and the individual reviews below, we regret to inform you that your work will not be considered further for publication in *eLife*.

In this nicely written manuscript, the authors show that shed *Plasmodium falciparum* EBA-175 mediates red cell clustering during parasite invasion and hence promotes parasite growth and also enables immune evasion. Although EBA-175 has been widely investigated and many aspects of this glycophorin binding protein are known, the function of the shed EBA-175 has not been defined. By performing a series of well-designed experiments, the authors expanded our understanding of this protein crucial for the survival of the parasite and a possible target for malaria vaccine.

The reviewers have also expressed major concerns and those are included in each individual review attached below. The main issue surrounds the lack of results in support of the authors claim concerning EBA-175-mediated clustering as a possible mechanism that may be occurring in vivo. The hypothesis that EBA-175 aggregated red blood cells may in fact contribute to pathogenesis in natural *Plasmodium falciparum* infection is indeed attractive. Therefore, the in vitro phenomenon should be replicated in vivo under physiological conditions and that may necessitate a substantial body of extra work and revision to validate the main hypothesis. In addition, validation of the purification of shed EBA-175, description and validation of flow cytometry data, and the number biological replicates needs to be included.

Should the authors perform the suggested experiments and provide data in support of their claims, they are encouraged to submit their manuscript as a new submission.

*Reviewer #1:*

The authors describe their observations concerning an interesting phenomenon, namely the shedding of EB-175 during *Plasmodium* infection of red blood cells. This shed EB-175 (actually a fragment of EBA-175 called RII is the main component that binds to GpA on RBC) not only potentiates the growth of the parasites by inducing agglutination or clustering of RBC, but it also causes or leads to immune evasion by preventing antibodies to access the parasite.

From my quick read of this manuscript I was quite excited about the EBA-175 story from Niraj Tolia's lab and thought the paper worthwhile for an external review. Upon closer reviewing and particularly examining the results and the approach, my enthusiasm has somewhat diminished. They support their claim of the shed EBA-175-induced clustering by showing that antibodies against EBA-175 do prevent this clustering, whereby in the absence of these antibodies, the parasites do expand/grow. This is shown in Figure 3 with Ab-217 that prevent parasite growth. However, I am not clear about this particular representation of their data mainly because nowhere in the manuscript do the authors provide adequate explanations about this particular manner of data presentation, stats, normalization, etc.

In essence, I have some concern about the veracity of some of the results given the number of biological replicates as in the case of results shown in Figure 1C, which represent a single biological replicate, of 5 technical replicates. Similarly, the results in Figure 2B that demonstrate the clustering effect brought about by RII, at titrated concentration, and not by F1 or F2, appear to be from a single biological replicate; the figure legend is not particularly informative. Similar concern is directed to results in Figure 2D where the authors show that treating Plasmodium RBC cultures with NA and with antibodies to GpA also inhibits clustering. It is not clear if the differences between untreated and treated are significant? Subsection “EBA-173f RII promotes RBC clusters in a GpA dependent manner” – the results are poorly described. In general, although the in vitro results are interesting, some results are difficult to understand.

The authors propose that this mechanism of EBA-175 shedding may actually occur in persons suffering from *P. falciparum* infection and that this shedding may lead to malaria-induced pathogenesis. Although an attractive hypothesis, the authors claim that it would be impossible to test it in humans because the shed EBA-175 could not be measured in sera of malaria patients as the protein would be bound to RBC. Could clustered RBC be detected in *P. falciparum* infected persons?

I would think that the authors could have tested this hypothesis in vivo studies in mice, e.g., mice with deleted GpA in which case the shed EBA-175 would not bind to RBC? Alternatively, would it be possible to have used CRISPR/Cas9 to edit the EBA-175 gene? Hence, although the results are potentially interesting and the suggestion that such mechanism may in fact operate to promote the spread of malaria infection, in my opinion, the results are not fully convincing.

*Reviewer #2:*

This manuscript by Paing et al. is a nicely written account of a series of well designed and simple experiments that show that clustering of red blood cells in in vitro culture is mediated by region 2 (RII) of EBA-175 shed by the parasite during invasion. The authors hypothesise that this clustering of non-infected red blood cells (RBCs) around infected RBCs (iRBCs) also occurs in natural infections of *P. falciparum*, and that it benefits the parasite by increasing replication rates through reducing the amount of time required for daughter merozoites to encounter new RBCs to invade, and through shielding of the iRBC from the action of growth and invasion inhibitory antibodies.

The experiments that show that clustering occurs in in vitro cultures, and that this can be mediated through shed RII of EBA-175, are on the surface well thought out and conducted, but could be made greatly more robust by validation of the purification of shed EBA-175, and description and validation of the flow cytometry data.

Moreover, the author's interpretation of the utility of this phenomenon to the parasite is based on the assumption that clustering also occurs in vivo. I think that the authors need to present evidence that 'clusters' are viable under the large stresses and shear forces encountered in the host blood stream, and that they remain viable under these conditions for the 48 hours required for the daughter merozoites to egress from the original iRBC. Apparatus and protocols for measuring the effects of flow on malaria parasite iRBCs are well established (e.g. flow chamber type experiments such as Dasanna et al., 2017), and would benefit this work immensely. I feel that without evidence that clustering could occur in vivo, the authors are overreaching in their interpretation of the significance of these results. I recommend that the authors test the robustness of these clusters under relevant physiological conditions in flow chamber experiments. There may also be scope to investigate clustering in rodent malaria parasites.

Introduction:

Clumping of erythrocytes by bivalent antibodies is well described and, indeed, has been the basis of erythrocyte typing. A dimer of shed EBA-175 might be hypothesized to confer a similar activity, based upon a bivalent structure and affinity for GpA. Although it has been presented many times in the literature, I think that a schematic of the domain structure of EBA-175, would introduce the general reader to the arrangement of RII, F1, F2, low complexity region, RVI, and the TM domain. Perhaps a model comparing the predicted structure and dimensions of bivalent shed EBA-175 with a bivalent antibody, to support the hypothesis of a clumping capacity.

The *Plasmodium* literature has a long history describing erythrocyte rosetting, this should also be introduced, and the parasite proteins proposed to be involved, and distinguished from a new hypothesis involving EBA-175. It would be useful for the reader to provide a thorough definition of 'clustering', as this term may be unfamiliar to many readers. Clustering should be compared to 'rosetting' and 'clumping' and their differences and similarities discussed.

*Plasmodium falciparum* is also known for the propensity of infection of erythrocytes by multiple ring forms, suggesting that the rupture of a schizont results in multiple events of an adjacent uninfected erythrocyte.

I am always wary when two distinct functions are attributed to a single parasite protein. Particularly given the propensity of *Plasmodium* to duplicate and amplify gene families. That is, for the parasite to take full advantage of evolutionary selective pressure for rosetting, it might duplicate the EBA-175 gene, use one copy specifically for invasion, and the second copy may evolve toward rosetting. Indeed, the DBL domains of specific PfEMP1 proteins are proposed to be involved in rosetting.

Results:

Figure 1A. I am stunned that the authors can immunoprecipitate sufficient shed EBA-175 to see on a Coomassie stained gel. How much culture was used in this experiment? My recollection is that the visualization methods in the literature are Western blotting or radioisotope labelling, is there precedence otherwise? The identity of the high MW band should be confirmed by Western blotting with an independent antibody, and since the material is so abundant by mass spec or peptide sequencing. As it stands, I am highly skeptical that it is possible to purify such abundant material.

Figures 1 and 2: The flow cytometry figure or legend should indicate the number of events sampled, it appears to vary. Many of the plots are blown out, is the frequency of the events so rare? Is the flow cytometry data consistent with the rosetting seen by light microscopy? In Figure 1B I don't understand which indicates clustering, forward or side scatter, and if the authors are arguing that both populations are indicative? For example, the pattern in Figure 1B is different than Figure 2B; and again in Figure 1D. Perhaps a flow cytometry machine which captures images of the particles could be used.

Figure 1: It appears that Ab-218 significantly reduces clustering, yet this antibody is known to target a non-functional epitope of EBA-175. What is the explanation for this?

Figure 2: Perhaps another region of EBA-175 should be used as a negative control here.

Figure 3: Can the authors propose a reason for the fact that the growth of Dd2 is enhanced to greater degree than the other strains? Does Dd2 grow more slowly than the other strains under normal culture conditions? Could it be that the merozoites of Dd2 have shorter viable life spans than merozoites of the other strains?

Figure 4: Again, another region of EBA-175 might make an appropriate negative control here.

Subsection “Shed EBA-175 purified from parasite culture induces RBC clustering”: "To identify a role for the shed protein…" This is overly presumptive, you should temper it by saying, "To identify if EBA-175 has an additional role after shedding…" Also in the Abstract, second sentence, qualify that a role is speculative.

Subsection “EBA-175 mediated RBC clustering is observed in parasite cultures”: "…nanomolar or greater levels." Please support, by reference or provide calculations.

Subsection “Clustering of RBCs mediated by RII enhances parasite growth”: It is unnecessary to emphasize "gold standard."

Subsection “Clustering of RBCs mediated by RII enhances parasite growth”: "Dd2 has previously been shown…" I don't understand the reasoning here, that the effect of exogenous EBA-175 is somehow correlated with Dd2's dependence on endogenous EBA-175. The clustering effect should be independent, no?

Subsection “Clustering of RBCs mediated by RII enhances parasite growth”: Is it necessary to hypothesize that RII might have some direct relationship with hypoxanthine incorporation, that must be controlled for with a second growth assay?

Subsection “Clustering of RBCs mediated by RII enhances parasite growth “: "…likely due to…" is speculation.

Subsection “Clustering of RBCs mediated by RII enhances parasite growth”, last paragraph: Do you mean to say Figure 3B?

Subsection “EBA-175-RII-mediated RBC clustering confers parasites protection from diverse neutralizing Abs”, first paragraph: I don't think that basigin needs to be capitalized.

Discussion:

In general, additional validation of the EBA-175 purification and flow cytometry experiments is required, and the Discussion is too far reaching in terms of the potential biological significance of this work. Without firm evidence that clustering occurs in vivo under typical physiological conditions, the speculation pertaining to the advantages effects of cluster formation is not justified based on the experimental evidence presented here. The authors have shown convincingly that EBA-175 mediated clustering occurs in in vitro cultures, and that this can lead to increased parasite growth rates, but their experiments do not show that this occurs in natural infections.

Discussion, first paragraph: 'uninfected' should be used instead of 'naïve'.

Discussion, second paragraph: The situation regarding EBL and virulence in *P. yoelii* is more complicated than the authors describe. The redirected localisation of EBL from the micronemes to the dense granules that occurs as a result of a mutation in R6 of EBL in 17XL, leads to a change in red blood cell tropism; from a predilection for the invasion of reticulocytes to the ability to invade RBCs of all ages. This change in RBC cell invasion preference cannot be explained through enhanced clustering. Furthermore, it has also been shown that another mutation in PyEBL, this time in RII, also increases virulence (again through an increase in the RBC invasion repertoire), but does not lead to localisation away from the micronemes (Abkallo et al., 2017).

References:

Abkallo HM, Martinello A, Inoue M, Xangsayarath P, Gitaka J, Yahata K, … Culleton R. Rapid identification of genes controlling virulence and immunity in malaria parasites. PLoS Pathog. 2017 Jul 12;13(7):e1006447. doi: 10.1371/journal.ppat.1006447

Dasanna AK, Lansche C, Lanzer M, Scwarz US. Rolling Adhesion of Schizont Stage Malaria-Infected Red Blood Cells in Shear Flow. Biophys J. 2017 May 9;112(9):1908-1919. doi: 10.1016/j.bpj.2017.04.001.

*Reviewer #3:*

EBA-175 plays an important role in binding to glycophorin A (GpA) during *Plasmodium falciparum* invasion of erythrocytes. However, EBA-175 is shed post invasion and a role for this shed protein has not been defined. The authors showed that EBA-175 shed from parasites promotes clustering of RBCs. Region II of EBA-175 is sufficient for clustering RBCs in a GpA-dependent manner. EBA-175-dependent RBC clustering enhances parasite growth and provides a general method to protect the invasion machinery from immune recognition. Finally, they proposed a mechanistic framework for the role of shed EBA-175 in RBC clustering, immune evasion, and severe malaria. I think the experiments were carefully designed and conducted; the results were appropriately discussed, and clearly written. I also agree them that these data will serve as a baseline information understanding novel roles of erythrocyte binding proteins.

1) Introduction

Please insert one paragraph which describes the reason(s) why they focused on the shed-EBA-175 and RBC clustering? Rosetting should also be mentioned in the Introduction.

2) Results (Subsection “EBA-175-RII-mediated RBC clustering confers parasites protection from diverse neutralizing Abs”, first paragraph)

Majority in this paragraph contains introductory description. Please bring this to Introduction section.

3) Figure 3D clearly indicated that a cluster contains 1 or 2 infected RBC. In this experimental condition, it is very important to know how much% of the RBC clusters contains iRBC? Are there any iRBC negative clusters? Also, it is important to know how much% of the iRBC associated with normal RBC clusters?

[Editors’ note: what now follows is the decision letter after the authors submitted for further consideration.]

Thank you for resubmitting your work entitled "Shed EBA-175 mediates red blood cell clustering that enhances malaria parasite growth and enables immune evasion" for further consideration at *eLife*. Your revised article has been favorably reviewed by three peer reviewers, one of whom is a member of our Board or Reviewing Editors, and the evaluation has been overseen by Tadatsugu Taniguchi as the Senior Editor.

The manuscript has been improved but there are some remaining issues that need to be addressed before acceptance. For the most part the suggested revisions concern clarifications of the experimental approaches, e.g., rounds of parasite replications, inclusions of more detailed legend explanations, or corrections of typographical errors. The comments from the reviewers are included below.

*Reviewer #1:*

The new submission of the revised manuscript by Niraj Tolia is now sufficiently improved. The majority of queries were addressed, particularly by providing results from additional experiments and clarifying entries in the legends and the text concerning biological replicates.

There are no substantive concerns regarding this current version of the revised manuscript.

*Reviewer #2:*

This manuscript is a revised re-submission of the previous version by Prof. Niraj H Tolia's group. Based on the previous reviewers' comments, the biggest concern was that the clustering phenomenon clearly showed in vitro should be proved under physiologically relevant flow conditions to establish the in vivo relevance of the findings. The authors conducted additional experiments to prove and answer to this critical question described in the Materials and methods section, "Cluster Formation Assays under Flow" and "Quantification of Cluster Under Flow", Figure 2E and Figure 2—figure supplement 2B.

I think all the experiments including additional flow experiments were carefully designed and conducted; the results were appropriately discussed, and clearly written. I also agree them that these data will serve as a baseline information understanding novel roles of erythrocyte binding proteins. However, I still have the following comments to improve this re-submitted manuscript.

1) Please clearly describe which experiments were conducted in the flow-condition in the respective figure legends.

"EBA-175 RII induced clusters are viable under physiological flow conditions" Please add data when no RII has added in Figure 2E.

2) "for two rounds of replication.…… erythrocytes in circulation."

Why did they use two rounds of replication (not one cycle assay)? Please describe the reason.

3) "Concentrations of RII greater than 100 nM … both DBL domains (Salinas and Tolia, 2014; Tolia et al., 2005; Wanaguru et al., 2013)." Should be in the Discussion section. But I felt this part sounds over discussion. Please consider to remove.

4) Subsection “EBA-175-RII-mediated RBC clustering confers parasites protection from diverse neutralizing Abs”, last paragraph: To interpret the results described here, the authors should pay more careful attention to the timings when RH5, AMA1 and EBA-175 works during merozoite invasion into the erythrocyte.

*Reviewer #3:*

In this eloquent manuscript, the authors investigate the phenomenon whereby EBA-175 shed by *Plasmodium* parasites during invasion promotes red blood cell clustering in culture. They show that the clustering is mediated by the glycophorin A binding region of EBA-175, called RII. Intriguingly, they observe that addition of recombinant RII protein to generate RBC clusters both enhances parasite growth and serves as an immune evasion strategy by hindering access of invasion-inhibitory antibodies.

The authors clearly demonstrate the EBA-175-mediated clustering of RBCs in vitro, and have revised an earlier version of the manuscript to include evidence that clustering occurs under conditions approximating physiological flow conditions. However, there is no in vivo evidence for EBA-175 mediated clustering, such as the observation of clusters from patients, or an estimate of the blood concentration of EBA-175 (which latter the authors posit is not possible). As a result, the authors' hypothesis that EBA-175-mediated clustering plays a role in enhancing parasite growth and immune evasion in severe malaria is speculative. Further, there is controversy surrounding the function of rosetting in vivo – as the authors point out in the Discussion, there is limited experimental evidence that rosetting leads to increased parasitemia or immune evasion. Nevertheless, I agree with the authors' comment that field studies are beyond the scope of this paper. In light of this, I would suggest that the authors make a point in the Discussion that their model still needs to be validated in vivo.

Overall I think this work is interesting and novel, and the experiments well-designed, well-executed, and well-presented.

---

## [Author Response]

[Editors’ note: the author responses to the first round of peer review follow.]

We appreciate the in-depth review and have revised the manuscript to address the reviewers’ concerns. The new version includes additional results to show that EBA-175 induced clusters are stable under physiologically relevant flow condition. We also present mass spectrometry data that unambiguously establishes the purified protein is shed native EBA-175. We have also significantly revised the manuscript to improve the description of flow cytometry data and experimental replicates.

[…] The reviewers have also expressed major concerns and those are included in each individual review attached below. The main issue surrounds the lack of results in support of the authors claim concerning EBA-175-mediated clustering as a possible mechanism that may be occurring in vivo. The hypothesis that EBA-175 aggregated red blood cells may in fact contribute to pathogenesis in natural Plasmodium falciparum infection is indeed attractive. Therefore, the in vitro phenomenon should be replicated in vivo under physiological conditions and that may necessitate a substantial body of extra work and revision to validate the main hypothesis. In addition, validation of the purification of shed EBA-175, description and validation of flow cytometry data, and the number biological replicates needs to be included.

We thank the reviewers for the in-depth review. We have provided additional experiments requested by the reviewers to study the clustering phenomenon under physiologically relevant flow conditions to establish the in vivo relevance of the findings. These are now presented in Figure 2 and Figure 2—figure supplement 2 as well as an independent section in the Results.

In addition, as requested by the reviewers, we now provide mass spectrometry data to unambiguously validate the purified protein is shed EBA-175. We have also extensively revised the manuscript to improve the presentation and description of replicates, and to improve the presentation of the flow cytometry data.

Should the authors perform the suggested experiments and provide data in support of their claims, they are encouraged to submit their manuscript as a new submission.Reviewer #1:[…] From my quick read of this manuscript I was quite excited about the EBA-175 story from Niraj Tolia's lab and thought the paper worthwhile for an external review. Upon closer reviewing and particularly examining the results and the approach, my enthusiasm has somewhat diminished. They support their claim of the shed FBA-175- induced clustering by showing that antibodies against EBA-175 do prevent this clustering, whereby in the absence of these antibodies, the parasites do expand/grow. This is shown in Figure 3 with Ab-217 that prevent parasite growth. However, I am not clear about this particular representation of their data mainly because nowhere in the manuscript do the authors provide adequate explanations about this particular manner of data presentation, stats, normalization, etc.In essence, I have some concern about the veracity of some of the results given the number of biological replicates as in the case of results shown in Figure 1C, which represent a single biological replicate, of 5 technical replicates. Similarly, the results in Figure 2B that demonstrate the clustering effect brought about by RII, at titrated concentration, and not by F1 or F2, appear to be from a single biological replicate; the figure legend is not particularly informative. Similar concern is directed to results in Figure 2D where the authors show that treating Plasmodium RBC cultures with NA and with antibodies to GpA also inhibits clustering. It is not clear if the differences between untreated and treated are significant? Subsection “EBA-173f RII promotes RBC clusters in a GpA dependent manner” – the results are poorly described. In general, although the in vitro results are interesting, some results are difficult to understand.

We apologize that the number of replicates for each experiment was not adequately presented. A thorough explanation of the number of technical and biological replicates in each experiment was added to the manuscript (in figure legends and in text). A minimum of three biological replicates were conducted for each experiment. For those experiments that were analyzed for statistics, at least five biological replicates were conducted.

The authors propose that this mechanism of EBA-175 shedding may actually occur in persons suffering from P. falciparum infection and that this shedding may lead to malaria-induced pathogenesis. Although an attractive hypothesis, the authors claim that it would be impossible to test it in humans because the shed EBA-175 could not be measured in sera of malaria patients as the protein would be bound to RBC. Could clustered RBC be detected in P. falciparum infected persons?

This would require setting up a field study which regrettably is beyond the scope of this manuscript.

I would think that the authors could have tested this hypothesis in vivo studies in mice, e.g., mice with deleted GpA in which case the shed EBA-175 would not bind to RBC? Alternatively, would it be possible to have used CRISPR/Cas9 to edit the EBA-175 gene? Hence, although the results are potentially interesting and the suggestion that such mechanism may in fact operate to promote the spread of malaria infection, in my opinion, the results are not fully convincing. However, I am going to defer my final opinion once we hear from the experts in the field.

Regrettably, these experiments are not possible. This is primarily because the EBA-175/GpA interaction is not replicated in mouse models of malaria. EBA-175 cannot bind murine glycophorin A. This lack of binding is due to a modification on the sialic acid of mouse Oglycan that is not found in humans. In addition, there is no direct ortholog of EBA-175 that can be studied in mice. A CRISPR/Cas9 knock out of EBA-175 would also not add to the manuscript as traditional knock outs of EBA-175 show little to no phenotype in culture. Knockout studies are also confounded by the redundancy in the EBL family of proteins in *P. falciparum*, and are unlikely to provide insight.

Reviewer #2:[…] The experiments that show that clustering occurs in in vitro cultures, and that this can be mediated through shed RII of EBA-175, are on the surface well thought out and conducted, but could be made greatly more robust by validation of the purification of shed EBA-175, and description and validation of the flow cytometry data.

We agree that a more robust validation of the purification of shed EBA-175 would improve the manuscript. In the revised manuscript we provide mass spectrometry data to establish that the purified protein is shed EBA-175. We also improve the description of the flow cytometry data to improve clarity.

Moreover, the author's interpretation of the utility of this phenomenon to the parasite is based on the assumption that clustering also occurs in vivo. I think that the authors need to present evidence that 'clusters' are viable under the large stresses and shear forces encountered in the host blood stream, and that they remain viable under these conditions for the 48 hours required for the daughter merozoites to egress from the original iRBC. Apparatus and protocols for measuring the effects of flow on malaria parasite iRBCs are well established (e.g. flow chamber type experiments such as Dasanna et al., 2017), and would benefit this work immensely. I feel that without evidence that clustering could occur in vivo, the authors are overreaching in their interpretation of the significance of these results. I recommend that the authors test the robustness of these clusters under relevant physiological conditions in flow chamber experiments. There may also be scope to investigate clustering in rodent malaria parasites.

We agree that demonstrating the clusters are stable under physiologically relevant shear forces would establish the in vivo relevance of the findings. In the revised manuscript we now show that EBA-175 mediated clusters are stable to the large stresses and shear forces induced by relevant physiological conditions in flow chamber experiments. The new data are included in Figure 2E and Figure 2—figure supplement 2B.

Regrettably, rodent models of malaria cannot be used to examine clustering by EBA-175 as mice modify sialic acid residues on glycophorin A, and this modification prevents binding by EBA175. Furthermore, a clear ortholog for EBA-175 has not been identified in rodent malaria parasites. This comment is related to our last response to reviewer 1.

Introduction:Clumping of erythrocytes by bivalent antibodies is well described and, indeed, has been the basis of erythrocyte typing. A dimer of shed EBA-175 might be hypothesized to confer a similar activity, based upon a bivalent structure and affinity for GpA. Although it has been presented many times in the literature, I think that a schematic of the domain structure of EBA-175, would introduce the general reader to the arrangement of RII, F1, F2, low complexity region, RVI, and the TM domain. Perhaps a model comparing the predicted structure and dimensions of bivalent shed EBA-175 with a bivalent antibody, to support the hypothesis of a clumping capacity.

A domain structure of EBA-175 II, including Ab-217 and Ab-218 binding sites highlighted, is now included in Figure 1—figure supplement 1A. We do not wish to speculate on a biophysical mechanism for clustering and as such we are reticent to add the requested comparison to bivalent antibody binding. While this might be one mechanism for how clustering could be achieved, there are multiple other methods for inducing clusters and we do not wish to bias readers without further biophysical evidence beyond the scope of this manuscript.

The Plasmodium literature has a long history describing erythrocyte rosetting, this should also be introduced, and the parasite proteins proposed to be involved, and distinguished from a new hypothesis involving EBA-175. It would be useful for the reader to provide a thorough definition of 'clustering', as this term may be unfamiliar to many readers. Clustering should be compared to 'rosetting' and 'clumping' and their differences and similarities discussed.

We use the term clustering/clumping of RBCs to distinguish it from rosetting where a single infected RBC with PfEMP1, STEVORs, or RIFINs on its surface is surrounded by uninfected RBCs. Clustering of RBCs described here is independent of these membrane-tethered proteins.

A discussion of the differences between rosetting and clustering is included in the revised manuscript: “The clustering phenomenon reported here exhibits similarities to *P. falciparum* rosetting attributed to the parasite membrane proteins PfEMP1 (Williams et al., 2012; Rowe et al., 1997; Moll et al., 2015), RIFINs (Chen et al., 1998) and STEVORs (Goel et al., 2015). …”

Plasmodium falciparum is also known for the propensity of infection of erythrocytes by multiple ring forms, suggesting that the rupture of a schizont results in multiple events of an adjacent uninfected erythrocyte.

The cultures were grown with shaking to minimize this occurrence as per published protocols. Furthermore, the microscopy images clearly show that daughter merozoites invade independent red cells in clusters in Figure 3D.

I am always wary when two distinct functions are attributed to a single parasite protein. Particularly given the propensity of Plasmodium to duplicate and amplify gene families. That is, for the parasite to take full advantage of evolutionary selective pressure for rosetting, it might duplicate the EBA-175 gene, use one copy specifically for invasion, and the second copy may evolve toward rosetting. Indeed, the DBL domains of specific PfEMP1 proteins are proposed to be involved in rosetting.

We agree that the reviewer that gene duplications result in redundancy and specialization within parasite families. We believe this might occur within the EBL family as well. This will always be a concern for any members of the EBL family due to redundancy.

Results:Figure 1A. I am stunned that the authors can immunoprecipitate sufficient shed EBA-175 to see on a Coomassie stained gel. How much culture was used in this experiment? My recollection is that the visualization methods in the literature are Western blotting or radioisotope labelling, is there precedence otherwise? The identity of the high MW band should be confirmed by Western blotting with an independent antibody, and since the material is so abundant by mass spec or peptide sequencing. As it stands, I am highly skeptical that it is possible to purify such abundant material.

Four liters of parasite culture was split into equal portions for pulldown with either the control antibody (Ab-8C2) or the EBA-175 specific antibody (Ab-218) following the procedure adapted from O’Donnell et al. (2006). Purified native protein was identified as EBA-175 by immunoblotting (new data – Figure 1—figure supplement 1B) and by mass spectrometry (new data – Figure 1—figure supplement 1C-D). These results clearly demonstrate that we have successfully purified high quality EBA-175 from parasite culture.

Figures 1 and 2: The flow cytometry figure or legend should indicate the number of events sampled, it appears to vary. Many of the plots are blown out, is the frequency of the events so rare? Is the flow cytometry data consistent with the rosetting seen by light microscopy? In Figure 1B I don't understand which indicates clustering, forward or side scatter, and if the authors are arguing that both populations are indicative? For example, the pattern in Figure 1B is different than Figure 2B; and again in Figure 1D. Perhaps a flow cytometry machine which captures images of the particles could be used.

We apologize for not being clear about the number of replicates conducted for each experiment. We have revised Figure 1 legend so that it now reads “(B)…Frequency of events (clusters) out of one million counts are located in bottom right corner of dot plot for one representative technical replicate… (D) … Frequency of events (clusters) out of one million counts are located in bottom right corner of dot plot for one representative technical replicate. …”

We have also revised Figure 2 legend so that it now reads “(B)…Frequency of events (clusters) out of 100,000 counts are located in bottom right corner of dot plot for each sample when gated according to the no protein control (top left). One representative biological replicate out of three is presented… (D)… Frequency of events (clusters) out of 100,000 counts are located in bottom right corner of dot plot for each sample when gated according to the no protein control (top left). One representative biological replicate out of three is presented…”

We also agree that a flow cytometry instrument with the ability to capture images would be a benefit although such an experiment would not add greatly to the manuscript. In lieu of additional flow cytometry data, we conducted experiments under physiological flow conditions and captured images of the clusters (Figure 2—figure supplement 2B).

Figure 1: It appears that Ab-218 significantly reduces clustering, yet this antibody is known to target a non-functional epitope of EBA-175. What is the explanation for this?

While significant compared to untreated, the inhibition by Ab-218 is slight and likely due to the amount of antibody used. Ab-218 has been shown to have slight inhibition of EBA-175 at high concentrations.

Figure 2: Perhaps another region of EBA-175 should be used as a negative control here.

We used both F1 and F2 as negative controls. The individual DBL domains serve as controls since either domain alone does not cluster RBCs. These are the most accurate negative control as they are expressed, refolded and purified in the same way as RII. These negative controls also demonstrate that an intact RII is required for the effect. It is unclear what additional negative controls will provide.

Figure 3: Can the authors propose a reason for the fact that the growth of Dd2 is enhanced to greater degree than the other strains? Does Dd2 grow more slowly than the other strains under normal culture conditions? Could it be that the merozoites of Dd2 have shorter viable life spans than merozoites of the other strains?

All strains used in experiments have between 45-48 hr life cycle, therefore the cycle time and life span is unlikely to contribute to the higher growth rate observed for Dd2. It is evident that the specific increase in growth by Dd2 is due to multiple factors and we do not wish to overly speculate in one direction or another. The exact mechanism for enhanced growth by Dd2 does not alter the main conclusions and focus of the manuscript that EBA-175 clustering enhances parasite growth and protects from neutralizing antibodies.

Figure 4: Again, another region of EBA-175 might make an appropriate negative control here.

F1 and F2 individual domains of RII have already shown not to induce enhanced parasite growth in Figure 3B. As described in our above response to reviewer 2, these serve as the most accurate negative controls.

Subsection “Shed EBA-175 purified from parasite culture induces RBC clustering”: "To identify a role for the shed protein…" This is overly presumptive, you should temper it by saying, "To identify if EBA-175 has an additional role after shedding…" Also in the Abstract, second sentence, qualify that a role is speculative.

We agree with the reviewer and have changed the sentence as requested.

Subsection “EBA-175 mediated RBC clustering is observed in parasite cultures”: "…nanomolar or greater levels." Please support, by reference or provide calculations.

We agree with this reviewer that this is speculative and have removed the sentence.

Subsection “Clustering of RBCs mediated by RII enhances parasite growth”: It is unnecessary to emphasize "gold standard."

We agree with the reviewer and have revised the sentence accordingly.

Subsection “Clustering of RBCs mediated by RII enhances parasite growth”: "Dd2 has previously been shown…" I don't understand the reasoning here, that the effect of exogenous EBA-175 is somehow correlated with Dd2's dependence on endogenous EBA-175. The clustering effect should be independent, no?

We agree with the reviewer that this sentence is not clear. We have removed it from the revised manuscript.

Subsection “Clustering of RBCs mediated by RII enhances parasite growth”: Is it necessary to hypothesize that RII might have some direct relationship with hypoxanthine incorporation, that must be controlled for with a second growth assay?

Using multiple assays ensures the results are robust and not artifacts of one particular assay. We are trying to be thorough to rule out the global effects ^3^H-labelled exogenous hypoxanthine could have on parasite growth.

Subsection “Clustering of RBCs mediated by RII enhances parasite growth”: "…likely due to…" is speculation.

We have revised the text to: “Concentrations of RII greater than 100 nM gradually caused parasite death. Plausible reasons for the parasite death seen are a block of invasion and/or due to formation of larger and more tightly-packed RBC clusters blocking…”

Subsection “Clustering of RBCs mediated by RII enhances parasite growth”, last paragraph: Do you mean to say Figure 3B?

We thank the reviewer for identifying this error. We have revised the reference to the correct figure.

Subsection “EBA-175-RII-mediated RBC clustering confers parasites protection from diverse neutralizing Abs”, first paragraph: I don't think that basigin needs to be capitalized.

We thank the reviewer for identifying this error. We have revised the sentence as requested.

Discussion:In general, additional validation of the EBA-175 purification and flow cytometry experiments is required, and the Discussion is too far reaching in terms of the potential biological significance of this work. Without firm evidence that clustering occurs in vivo under typical physiological conditions, the speculation pertaining to the advantages effects of cluster formation is not justified based on the experimental evidence presented here. The authors have shown convincingly that EBA-175 mediated clustering occurs in in vitro cultures, and that this can lead to increased parasite growth rates, but their experiments do not show that this occurs in natural infections.

We agree with the reviewer that examining clustering under physiologically relevant flow conditions establishes the in vivo role for this phenomenon. In the revised manuscript we present new clustering data under physiological flow conditions that demonstrate the EBA-175 clusters are stable under physiological stresses.

Discussion, first paragraph: 'uninfected' should be used instead of 'naïve'.

We agree with this change and have revised the manuscript where appropriate.

Discussion, second paragraph: The situation regarding EBL and virulence in P. yoelii is more complicated than the authors describe. The redirected localisation of EBL from the micronemes to the dense granules that occurs as a result of a mutation in R6 of EBL in 17XL, leads to a change in red blood cell tropism; from a predilection for the invasion of reticulocytes to the ability to invade RBCs of all ages. This change in RBC cell invasion preference cannot be explained through enhanced clustering. Furthermore, it has also been shown that another mutation in PyEBL, this time in RII, also increases virulence (again through an increase in the RBC invasion repertoire), but does not lead to localisation away from the micronemes (Abkallo et al., 2017).

We agree with the reviewer that the relationship of clustering to virulence in *P. yoelii* is complex. We have removed this section as requested.

Reviewer #3:[…]1) IntroductionPlease insert one paragraph which describes the reason(s) why they focused on the shed-EBA-175 and RBC clustering? Rosetting should also be mentioned in the Introduction.

We have added the following to the Introduction to explain the rationale for focusing on this phenomenon:

“In this study, we establish a role for shed EBA-175 in RBC clustering and examine the biological consequences of this phenomenon. We embarked on this line of investigation after observing potent clustering of red blood cells upon addition of the binding domain of EBA-175.”

We retained the description of rosetting in the Discussion. We feel this is easier for the reader to understand the relevance of rosetting and relationship to clustering once the results have been presented. Moving the description of rosetting to the Introduction does not fit with the focus of the manuscript (i.e. EBA-175) and is confusing as EBA-175 has no role in rosetting.

2) Results (Subsection “EBA-175-RII-mediated RBC clustering confers parasites protection from diverse neutralizing Abs”, first paragraph)Majority in this paragraph contains introductory description. Please bring this to Introduction section.

We agree and this material was moved to the Introduction.

3) Figure 3D clearly indicated that a cluster contains 1 or 2 infected RBC. In this experimental condition, it is very important to know how much% of the RBC clusters contains iRBC? Are there any iRBC negative clusters? Also it is important to know how much% of the iRBC associated with normal RBC clusters?

In the culture, both clusters with and without iRBC can be seen. Clusters on average contain 1 to 2 iRBCs although as many as 7 could be seen in a cluster. Not all iRBC can be seen at once as the cluster is a three dimensional structure and would require the use of z-stacks to image every iRBC and uRBC in the cluster. This was not done for this experiment although multiple images were taken of the larger clusters like the one in Figure 3D to observe other iRBCs.

While we agree with the reviewer that establishing the number and composition of clusters is an important future aspect for this research, we feel these results are beyond the scope of the current manuscript. In addition, while these results would provide interesting insight they do not add to the current focus of the manuscript which describes the novel phenomenon of clustering.

[Editors' note: the author responses to the re-review follow.]

Reviewer #2:[…] 1) Please clearly describe which experiments were conducted in the flow-condition in the respective figure legends.

We apologize for any confusion in this figure legend. The only experiment in Figure 2 that was conducted under physiological flow conditions was (E). We have revised the legend for Figure 2E to read “Analysis of cluster size for varying concentrations of EBA-175 RII under flow conditions of 1dyn/cm^2^.”

"EBA-175 RII induced clusters are viable under physiological flow conditions" Please add data when no RII has added in Figure 2E.

The data in Figure 2E comprises the µm^2^ values for all observed clusters formed during flow assay experiments, and calculated from the captured images. In the absence of RII, no clusters were observed in flow, as such there were no data to record.

2) "for two rounds of replication.… erythrocytes in circulation."Why did they use two rounds of replication (not one cycle assay)? Please describe the reason.

We apologize to the reviewer that this was not clear. The growth enhancement effect by RII was detectable after one round, but we used two rounds of replication to achieve minimal noise in the assay. We changed the lines in question to now read “…We assessed whether clustering of RBCs had any effects on parasite growth in culture using diverse laboratory strains to identify strain-transcending effects. *[…]* The proliferation and/or growth inhibition of the parasites was assessed using the approach of analyzing the incorporation of [^3^H]-hypoxanthine into parasite nucleic acids after two rounds of replication to get a better signal-to-noise ratio…”

3) "Concentrations of RII greater than 100 nM … both DBL domains (Salinas and Tolia, 2014; Tolia et al., 2005; Wanaguru et al., 2013)." Should be in the Discussion section. But I felt this part sounds over discussion. Please consider to remove.

We agree with the reviewer and have removed this section.

4) Subsection “EBA-175-RII-mediated RBC clustering confers parasites protection from diverse neutralizing Abs”, last paragraph: To interpret the results described here, the authors should pay more careful attention to the timings when RH5, AMA1 and EBA-175 works during merozoite invasion into the erythrocyte.

We apologize to the reviewer if this section was not clear. We are assessing the advantage of clustering on subsequent rounds of invasion which would follow EBA-175 cleavage and binding back to RBCs forming clusters. To make this more clear we have changed this section to read as follows “…We asked whether RBC clustering confers parasites a survival advantage, other than growth enhancement, in the presence of known neutralizing Abs during subsequent rounds of invasion following cluster formation. We assessed the effect of RII-mediated red cell clustering on parasite viability…”

Reviewer #3:[…] The authors clearly demonstrate the EBA-175-mediated clustering of RBCs in vitro, and have revised an earlier version of the manuscript to include evidence that clustering occurs under conditions approximating physiological flow conditions. However, there is no in vivo evidence for EBA-175 mediated clustering, such as the observation of clusters from patients, or an estimate of the blood concentration of EBA-175 (which latter the authors posit is not possible). As a result, the authors' hypothesis that EBA-175-mediated clustering plays a role in enhancing parasite growth and immune evasion in severe malaria is speculative. Further, there is controversy surrounding the function of rosetting in vivo – as the authors point out in the Discussion, there is limited experimental evidence that rosetting leads to increased parasitemia or immune evasion. Nevertheless, I agree with the authors' comment that field studies are beyond the scope of this paper. In light of this, I would suggest that the authors make a point in the Discussion that their model still needs to be validated in vivo.

We thank the reviewer for their response and have included the following sentence in the Discussion: “The model and data presented here forms a strong foundation for the future study of clustering and its role in immune evasion in patient populations.”